# Adaptation is maintained by the parliament of genes

Thomas W. Scott [ID] [1]* & Stuart A. West[1]

Fields such as behavioural and evolutionary ecology are built on the assumption that natural selection leads to organisms that behave as if they are trying to maximise their fitness. However, there is considerable evidence for selfish genetic elements that change the behaviour of individuals to increase their own transmission. How can we reconcile this contradiction? Here we show that: (1) when selfish genetic elements have a greater impact at the individual level, they are more likely to be suppressed, and suppression spreads more quickly; (2) selection on selfish genetic elements leads them towards a greater impact at the individual level, making them more likely to be suppressed; (3) the majority interest within the genome generally prevails over 'cabals' of a few genes, irrespective of genome size, mutation rate and the sophistication of trait distorters. Overall, our results suggest that even when there is the potential for considerable genetic conflict, this will often have negligible impact at the individual level.

---

[1] Department of Zoology, University of Oxford, Zoology Research and Administration Building, 11a Mansfield Road, Oxford OX1 3SZ, UK.
*email: thomas.scott@zoo.ox.ac.uk

There is a contradiction between major branches of modern evolutionary biology. On the one hand, fields such as behavioural and evolutionary ecology are based on the assumption that organisms will behave as if they are trying to maximise their fitness[1–4]. Models based on fitness maximisation are used to make predictions about the selective forces (reasons) for adaptation, and these are then tested empirically[5,6]. This approach has been phenomenally successful, explaining many aspects of behaviour, life history and morphology. For example, fitness maximisation underpins our evolutionary explanations of: foraging behaviour, resource competition, sexual selection, parental care, sex allocation, signalling and cooperation[7–12].

On the other hand, there is considerable evidence for selfish genetic elements, which increase their own contribution to future generations at the expense of other genes in the same organism[13–17]. These selfish genetic elements may distort traits away from the values that would maximise individual fitness, to increase their own transmission[14,18–22]. Evidence for such genetic conflict has been found across the tree of life, from simple prokaryotes to complex animals. The contradiction is that selfish genetic elements mess up individual fitness maximisation, and appear to be common, but individual fitness maximisation still appears to occur[17,23,24]. This contradiction is especially apparent in the study of sex allocation: theoretical models based on individual fitness maximisation have explained a wide range of natural variation in sex ratio, and yet there have been many reported cases of selfish sex ratio distorters[9,14,25–27].

Leigh[28] provided a potential solution to this contradiction by suggesting that selfish genetic elements would be suppressed by the 'parliament of genes'. Leigh's argument was that, because selfish genetic elements reduce the fitness of most of the other genes in the organism, these other genes will have a united interest in suppressing selfish genetic elements. Furthermore, because these other genes are far more numerous, they will be likely to win the conflict. Consequently, even when there is considerable potential for conflict within individuals, we would still expect fitness maximisation at the individual level[29–34]. Leigh[28] demonstrated the plausibility of his argument by showing theoretically how a suppressor of a sex ratio distorter could be favoured. Since then, numerous suppressors have been studied from a theoretical and an empirical perspective[14,35,36].

However, several issues may affect the validity of the parliament of genes hypothesis. First, whether a suppressor spreads can depend upon biological details such as the extent to which a selfish genetic element is distorting a trait, the population frequency of that element and the cost of suppression[14,37–43]. Are certain types of selfish genetic elements, which cause substantial distortion, less likely to be suppressed? Second, if the spread of suppressors through populations is slow, and if selfish genetic elements arise continuously over evolutionary time, nonequilibrium trait distortion may be possible[35]. Third, selfish genetic elements are themselves also under evolutionary pressure to cause a level of trait distortion that would maximise their transmission to the next generation[15]. Could the evolution of selfish genetic elements lead to trait distortion that is less likely to be suppressed?[32] Fourth, if a suppressor does not reach fixation in a population, or a selfish genetic element is not purged from a population, subsequent mating may decouple selfish genetic elements and suppressors to expose previously suppressed trait distortion[38]. How important is this problem of polymorphism likely to be?

We address these issues, by investigating the parliament of genes hypothesis theoretically. Our aim is to investigate the extent to which genetic conflict distorts traits away from the value that would maximise individual fitness. We find that: (i) the greater the level of trait distortion caused by a selfish genetic element, the

more likely and the quicker it is suppressed; (ii) selection on selfish genetic elements leads towards greater trait distortion, making them more likely to be suppressed; (iii) in genome-wide arms races to gain control of organism traits, the majority interest within the genome generally prevails over 'cabals of a few', regardless of genome size, mutation rate, and the strength and sophistication of trait distorters. We find the same patterns with an illustrative model, and when examining three specific scenarios: selfish trait distortion of the sex ratio by an X chromosome driver; an altruistic helping behaviour encoded by an imprinted gene; and production of a cooperative public good encoded on a horizontally transmitted bacterial plasmid. Furthermore, we find close agreement when analysing scenarios with population genetic analyses and individual-based simulations. Our results suggest that even when there is potential for considerable genetic conflict, it has relatively little impact on traits at the individual level.

## Results

**Modelling approach.** We examine conflict between two groups of genes within the genome. We assume a selfish genetic element that can gain a propagation advantage through distorting some trait of the organism ('trait distorter'). This trait distortion only benefits alleles at a subset of loci within the genome—Leigh termed this subset of loci a 'cabal'[30]. The rest of the genome, which does not gain the propagation advantage from the trait distortion, will be selected to suppress the trait distorter. Leigh termed this collection of genes, which will comprise most of the genome, and so will constitute the majority within the parliament of genes, the 'commonwealth'[30].

We used two complementary theoretical approaches. First, we developed 'Equilibrium models', where we assume that the trait distorter and their cabal are only a very small fraction of the genome. We allow for this by assuming that it is highly likely that a potential suppressor of a trait distorter can arise by mutation. Consequently, in these models, we focus our analyses on when a trait distorter and its suppressor can spread. We use this approach to examine, given the potential for suppression, what direction would we expect natural selection to take on average.

We then developed 'Dynamics models', where we relaxed the assumption that the trait distorter and its cabal are a negligible fraction of the genome. In this case, rather than focus on the equilibrium state, we allowed trait distorters and their suppressors to arise continuously, at different loci across the genome. This approach allows us to investigate the influence of factors such as genome size, mutation rate and cabal size. We use this approach to determine the outcome of an evolutionary conflict that embroils the whole genome, to elucidate how far an organism trait is likely to be distorted at any given point in evolutionary time.

**Equilibrium models.** We assessed, given the potential for suppression, the extent to which a trait distorter will distort an organism trait away from the optimum for individuals. In order to elucidate the selective forces, we ask four questions in a stepwise manner, with increasing complexity:

(1) In the absence of a suppressor, when can a trait distorter invade?

(2) When can a costly suppressor of the trait distorter invade?

(3) What are the overall consequences of trait distorter-suppressor dynamics for trait values, at the individual and population level, at evolutionary equilibrium and before equilibrium has been reached?

(4) If the extent to which the trait distorter manipulates the organism trait can evolve, how will this influence the

likelihood that it is suppressed, and hence the individual and population trait values?

We assume an arbitrary trait that influences organism fitness. In the absence of trait distorters, all individuals have the trait value that maximises their individual fitness. The trait distorter manipulates the trait away from the individual optimum, to increase their own transmission to offspring. We assume a large population of diploid, randomly mating individuals. The aim of this model is to establish key aspects of the population genetics governing trait distorters and their suppressors, in an abstract setting. In Supplementary Notes 3, 4 and 5, we address the same issues in three specific biological scenarios.

(1) Spread of a trait distorter: We consider a trait distorter, which we denote by $D_1$, that is dominant and distorts an organism trait value by some positive amount $k$ ($k > 0$). This trait distortion increases the transmission of the trait distorter to offspring. Specifically, the trait distorter ($D_1$) drives at meiosis, in heterozygotes, against a trait non-distorter ($D_0$), being passed into the proportion $(1 + t(k))/2$ of offspring. $t(k)$ denotes the transmission bias ($0 \leq t(k) \leq 1$) and is a monotonically increasing function of trait distortion $\left(\frac{dt}{dk} \geq 0\right)$.

We emphasise that, in nature, trait distorters need not be meiotic drivers—the key point here is that we are considering when trait distortion increases the propagation of that trait distorter. We chose meiotic drive in this model for simplicity, and model different mechanisms in the biologically specific models (Supplementary Notes 3, 4 and 5). Indeed, in many natural cases, meiotic drivers would not gain their advantage by distorting a trait, in which case they would not enter any conflict with the rest of the genome over organism trait values, and therefore would not have any lasting influence on whether trait values are those that maximise individual fitness. For example, the segregation distorter (SD) meiotic driver in *Drosophila melanogaster* gains its advantage in heterozygous males by disrupting the proper development of rival sperm, and not by trait distortion[44]. Any organism-level fitness costs associated with SD would be opposed by SD as well as across the rest of the genome[45]. Our focus in this paper is on selfish genetic elements that gain an advantage by trait distortion, and therefore disagree with the majority of genes over trait values.

Trait distortion leads to a fitness (viability) cost ($c_{trait}(k)$) at the individual level, reducing an individual's number of offspring from 1 to $1 - c_{trait}(k)$ ($0 \leq c_{trait}(k) \leq 1$). Owing to trait distorter dominance, the fitness cost of trait distortion is borne by heterozygous as well as trait distorter-homozygous individuals. The fitness cost is a monotonically increasing function of trait distortion $\left(\frac{dc_{trait}}{dk} \geq 0\right)$. We assume that $t(k)$ and $c_{trait}(k)$ do not change with population allele frequencies, but relax this assumption in our specific models.

We first ask what frequency the trait distorter will reach in the population in the absence of suppression. If we take $p$ and $p'$ as the population frequency of the trait distorter in two consecutive generations, then the population frequency of the trait distorter in the latter generation is:

$$\bar{w} p' = (1 - c_{trait}(k)) \left(p^2 + (1-p)p(t(k)+1)\right), \quad (1)$$

where $\bar{w}$ is the average fitness of individuals in the population in the current generation, and can be written in full as: $\bar{w} = (1 - c_{trait}(k))(p^2 + 2p(1-p)) + (1-p)^2$. In 'Trait distorter population frequency' in the Methods, we show, with a population genetic analysis of Eq. 1, that the trait distorter will spread from rarity and reach fixation when $c_{trait}(k) < t(k)(1 - c_{trait}(k))$. This shows that trait distortion will evolve when the number of offspring that the trait distorter gains as a result of trait distortion ($t(k)(1 -$

$c_{trait}(k))$) is greater than the number of offspring bearing the trait distorter that are lost as a result of reduced individual fitness ($c_{trait}(k)$).

(2) Spread of an autosomal suppressor: We assume that the trait distorter ($D_1$) can be suppressed by an unlinked autosomal allele (suppressor), denoted by $S_1$. We assume that this suppressor ($S_1$) is dominant and only expressed in the presence of the trait distorter (facultative), but found similar results when the suppressor is constitutively expressed (obligate; Supplementary Note 6). Expression of the suppressor incurs a fitness cost to the individual, $c_{sup}$ ($0 \leq c_{sup} \leq 1$), which could arise for multiple reasons, including energy expenditure, or errors relating to the use of gene silencing machinery[46,47]. Gene silencing generally precedes the translation of the targeted gene, and so we assume that the cost of suppression ($c_{sup}$) is independent of the amount of trait distortion caused by the trait distorter ($k$).

We can write recursions detailing the generational change in the frequencies of the four possible gametes, $D_0/S_0$, $D_0/S_1$, $D_1/S_0$ and $D_1/S_1$, with the respective frequencies in the current generation denoted by $x_{00}$, $x_{01}$, $x_{10}$ and $x_{11}$, and the frequencies in the subsequent generation denoted by an appended dash ('):

$$\bar{w} x'_{00} = x_{00}^2 + x_{00}x_{01} + (1-t)(1-c_{trait})x_{00}x_{10} \\ + \left((1-c_{sup})/2\right)x_{00}x_{11} + \left((1-c_{sup})/2\right)x_{01}x_{10}, \quad (2)$$

$$\bar{w} x'_{01} = x_{00}x_{01} + \left((1-c_{sup})/2\right)x_{00}x_{11} + x_{01}^2 \\ + \left((1-c_{sup})/2\right)x_{01}x_{10} + \left(1-c_{sup}\right)x_{01}x_{11}, \quad (3)$$

$$\bar{w} x'_{10} = (1+t)(1-c_{trait})x_{00}x_{10} + \left((1-c_{sup})/2\right)x_{00}x_{11} \\ + \left((1-c_{sup})/2\right)x_{01}x_{10} + (1-c_{trait})x_{10}^2 + \left(1-c_{sup}\right)x_{10}x_{11}, \quad (4)$$

$$\bar{w} x'_{11} = \left((1-c_{sup})/2\right)x_{00}x_{11} + \left((1-c_{sup})/2\right) \\ x_{01}x_{10} + \left(1-c_{sup}\right)x_{01}x_{11} + \left(1-c_{sup}\right)x_{10}x_{11} + \left(1-c_{sup}\right)x_{11}^2, \quad (5)$$

where $\bar{w}$ is the average fitness of individuals in the current generation, and equals the sum of the equations' right-hand sides. In 'Suppressor invasion condition' in the Methods, we show, with a population genetic analysis of these equations, that a suppressor will spread from rarity if trait distortion ($k$) is greater than some threshold value, at which the cost of suppression ($c_{sup}$) is less than the cost of being subjected to trait distortion, $c_{sup} < c_{trait}(k)$. A threshold with respect to the level of trait distortion ($k$) arises because the cost of trait distortion ($c_{trait}(k)$) increases with greater trait distortion, but the cost of suppression ($c_{sup}$) is constant. Given that the individual cost of pre-translational suppression at a single locus is likely to be low[46,47], trait distortion conferred by unsuppressed trait distorters is likely to be negligible.

(3) Consequences for organism trait values: The extent of trait distortion at the individual level shows a discontinuous relationship with the strength of the trait distorter (Fig. 1a). When trait distortion is low, a suppressor will not spread ($c_{sup} > c_{trait}(k)$) and so the level of trait distortion at the individual level will increase with the level of trait distortion induced by the trait distorter ($k$). However, once a threshold is reached ($c_{sup} < c_{trait}(k)$), the suppressor spreads. We show in 'Equilibrium trait distorter and suppressor frequencies' in the Methods that the spread of the suppressor ($S_1$) causes the trait distorter ($D_1$) to lose its selective advantage and be eliminated from the population, leading to an absence of trait distortion at the individual level. In contrast, we

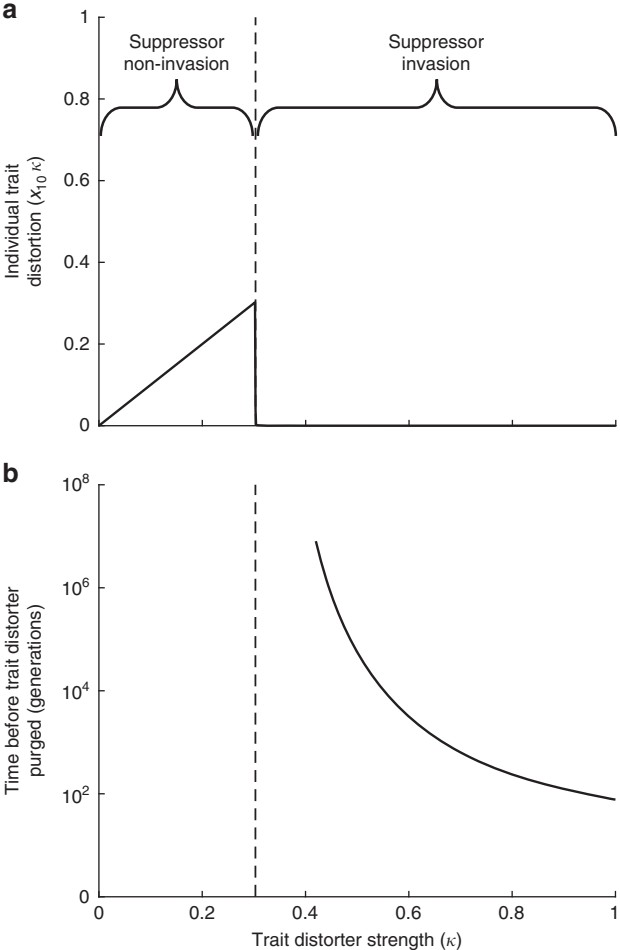

**Fig. 1** Trait distorter-suppressor dynamics and consequences for the organism. The trait distorter ($D_1$) and its suppressor ($S_1$) are introduced from rarity. In **a**, the resulting average trait distortion ($x_{10} \, k$) is plotted at equilibrium, against the extent to which the trait distorter causes trait values to deviate from the individual optimum ($k$). Below a certain threshold strength ($c_{sup} > c_{trait}(k)$; left of dashed line), the suppressor does not invade, and so the resulting trait distortion increases with the strength of the trait distorter ($k$). Above this threshold, the suppressor invades, and the trait distorter is purged, restoring the trait to the individual optimum. In **b**, the number of generations between trait distorter introduction and loss is plotted, on a $\log_{10}$ scale, for trait distorters that are purged at equilibrium (having been suppressed), which lie to the right of the dashed line. Stronger trait distorters are purged more quickly than weaker trait distorters ($c_{sup} = 0.15$; $t = 0.87k$ and $c_{trait} = 0.9k^{1.5}$)

show in Supplementary Note 6 that if the suppressor is constitutively expressed (obligate), the spread of the suppressor ($S_1$) to fixation in the population causes the trait distorter ($D_1$) to become neutral, meaning the trait distorter ($D_1$) can be maintained in the population without being expressed.

Overall, these results suggest that, given a relatively low cost of suppression ($c_{sup}$), the level of trait distortion observed at the individual level will either be low or absent. When a trait distorter is weak (low $k$), it will not be suppressed, but it will only have a small influence at the level of the individual. When a trait distorter is strong (high $k$), it will be suppressed and so there will be no influence at the level of the individual (Fig. 1a).

In addition, we found that stronger trait distorters are suppressed more quickly (Fig. 1b). In 'Non-equilibrium trait distortion' in the Methods, we numerically iterated our recursions

to determine how many generations it takes for suppressors to reach equilibrium. As long as trait distortion continues to reduce individual fitness non-negligibly after suppression is favoured (such that $\frac{dt}{dk} / \frac{dc_{trait}}{dk}$ is not excessively high after $c_{sup} < c_{trait}(k)$), stronger trait distorters (higher $k$) are suppressed and purged more rapidly than weaker trait distorters, limiting the potential for non-equilibrium trait distortion (Fig. 1b).

(4) Evolution of trait distortion: We then considered the consequence of allowing the level of trait distortion ($k$) to evolve. We assume a trait distorter ($D_1$) that distorts by $k$, and then introduce a rare mutant ($D_2$) that distorts by a different amount $\hat{k}$ ($\hat{k} \neq k$). This mutant ($D_2$) is propagated into the proportion $(1 + t(\hat{k}) - t(k))/2$ of the offspring of $D_2D_1$ heterozygotes, and into the proportion $(1 + t(\hat{k}))/2$ of the offspring of $D_2D_0$ heterozygotes. We assume that the stronger of the two trait distorters is dominant, but found similar results when assuming additivity ('Invasion of a mutant trait distorter' in the Methods). We assume that the similarity in coding sequence and regulatory control means that the original trait distorter and the mutant are both suppressed by the same suppressor allele, at the same cost ($c_{sup}$)[46,47]. In 'Invasion of a mutant trait distorter' in the Methods, we write the recursions that detail the generational frequency changes in the different possible gametes ($D_0/S_0$, $D_0/S_1$, $D_1/S_0$, $D_1/S_1$, $D_2/S_0$ and $D_2/S_1$).

We found that stronger mutant trait distorters ($\hat{k} > k$) will invade from rarity when the marginal increase in offspring they are propagated into exceeds the marginal increase in offspring they are lost from as a result of reduced fitness ($\Delta t(1 - c_{trait}(\hat{k})) > \Delta c_{trait}$, where $\Delta$ denotes marginal change ($\Delta t = t(\hat{k}) - t(k)$; $\Delta c_{trait} = c_{trait}(\hat{k}) - c_{trait}(k)$)). Consequently, if trait distortion is initially low, and successive mutant trait distorters are introduced, each deviating only slightly from the trait distorters from which they are derived ('δ-weak selection'[48]), invading trait distorters will approach a 'target' strength, denoted by $k_{target}$. This target strength corresponds to the level of trait distortion that would maximise the fitness of the gene[15], and is when the marginal benefit of transmission is exactly counterbalanced by the marginal individual cost of reduced offspring, $\frac{dt}{dk}(1 - c_{trait}) = \frac{dc_{trait}}{dk}$. The target strength of trait distortion ($k_{target}$) will therefore be greater if increased trait distortion ($k$) leads to a low rate of decrease in marginal transmission benefit $\left(-\frac{d^2t}{dk^2}\right)$ relative to the rate of increase in marginal individual cost $\left(\frac{d^2c_{trait}}{dk^2}\right)$ (Fig. 2b). If mutations are larger (strong selection), invading trait distorters may overshoot the target strength of trait distortion ($\hat{k} > k_{target}$). Weaker mutant trait distorters ($\hat{k} < k$) are recessive so cannot invade from rarity.

As evolution on the trait distorter increases the level of trait distortion, it makes it more likely that the trait distorter goes above the critical level of trait distortion where suppression will be favoured. When this is the case ($c_{sup} < c_{trait}(k_{target})$), the trait distorter spreads to high frequency, which then causes the suppressor to increase in frequency, reversing the direction of selection on the trait distorter, towards non-trait distortion ($D_0$), resulting in 0 trait distortion at equilibrium ($k^* = 0$) (Fig. 2a; 'Equilibrium allele frequencies after mutant invasion' in the Methods). Suppression only fails to spread if the individual fitness cost associated with suppression is greater than the individual fitness cost associated with the target trait distortion ($c_{sup} > c_{trait}(k_{target})$; Fig. 2a). Given that the individual fitness cost of pre-translational suppression at a single locus is likely to be low, then any non-negligible trait distorter is likely to be suppressed.

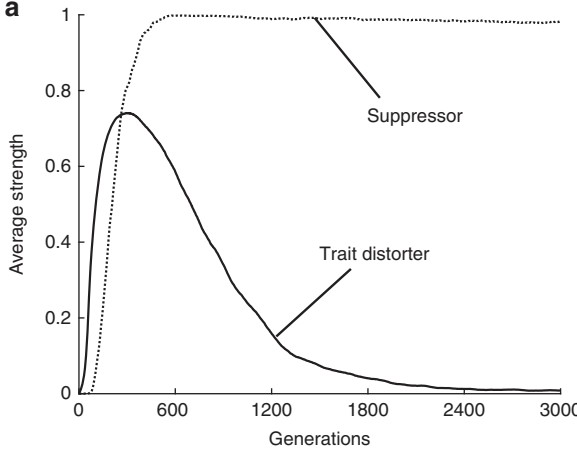

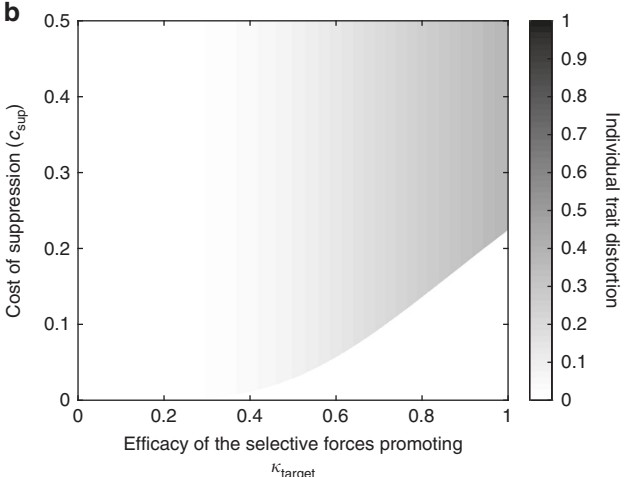

**Fig. 2** Evolution of trait distortion. In **a**, a trait distorter and suppressor are introduced in our agent-based simulation model (Methods: 'Agent-based simulation (single trait distorter locus)'), with $c_{sup} = 0.1$, $t = k$ and $c_{trait} = \max(k_a, k_b)/2$. The population average trait distorter and suppressor strengths over 100 simulation runs are plotted for successive generations. Initially, both trait distorter and suppressor strength increases, but then the trait distorters are purged from the population. **b** shows how the trait distortion at equilibrium is influenced by the cost of suppression ($c_{sup}$), and the target level of trait distortion ($k_{target}$), which is determined by the rate of increase in the marginal individual cost of trait distortion $\left(\frac{d^2 c_{trait}}{dk^2}\right)$ relative to the rate of decrease in the marginal transmission benefit $\left(-\frac{d^2 t}{dk^2}\right)$ (Supplementary Note 1). Trait distortion is low, unless there is both a high target level of trait distortion and a relatively high cost of suppression (top right of heat map)

Overall, our results suggest that selection on trait distorters will tend to lead to the eventual suppression of those trait distorters. In 'Agent-based simulation (single trait distorter locus)' in the Methods, we developed an agent-based simulation, which allowed us to continuously vary the level of both trait distortion and suppression, and obtained results in close agreement (Fig. 2a; Supplementary Note 2, Supplementary Fig. 2).

**Specific biological scenarios.** In Supplementary Notes 3, 4 and 5, we tested the robustness of our above conclusions by developing models for three different biological scenarios: a sex ratio distorter on an X chromosome (X driver); an imprinted gene that is only expressed when maternally inherited; and a gene for the

production of a public good by bacteria, which is encoded on a mobile genetic element[14,26,36,49–52]. We examined these cases because they are different types of trait distortion, involving different selection pressures, in very different organisms. In all three specific models, we obtained the same qualitative results as with our above illustrative model for an arbitrary trait (Fig. 3).

**Dynamics models.** Our Equilibrium models assumed that the suppressor of any given trait distorter will arise quickly by mutation. This assumption becomes less likely if suppressors are complex and hard to evolve, or favoured across a reduced portion of the genome (smaller commonwealth). Also, multiple trait distorters and their suppressors may arise continually in populations, through evolutionary time, at different loci within the cabal and commonwealth respectively. Organisms may therefore never rest at equilibria where all trait distorters are suppressed or of negligible strength.

We address these issues by relaxing our assumption that the commonwealth is very large relative to the cabal, assuming instead that the commonwealth encompasses some majority of loci within the genome, with the remaining loci comprising the cabal. We examined the average and extremes of trait distortion produced by trait distorters and suppressors, by asking three further questions, of increasing complexity, in a step-wise manner:

(5) To what extent are organism traits distorted when populations of individuals are only ever subjected to one segregating trait distorter at a time (no trait distorter co-segregation)?
(6) To what extent are organism traits distorted when populations of individuals may be exposed to multiple, co-segregating, interacting trait distorters?
(7) To what extent are organism traits distorted when the strength of each trait distorter may evolve?

(5) Trait distortion when no trait distorter co-segregation: We model a population of individuals, each with a genome size of $\gamma$ loci. Within this genome, the cabal constitutes a fraction $\theta$ of all loci, and the commonwealth constitutes the remaining fraction $1 - \theta$ of all loci. If a fraction of the genome is inherited in the same way, such that it favours the same trait values (same maximand), it is termed a 'coreplicon'[20,22]. The cabal comprises all coreplicons that favour the distortion of a particular trait, along a particular axis, in a particular direction, away from individual fitness maximisation. The commonwealth comprises the remaining replicons. Cabals and commonwealths are therefore trait-specific. It is useful, when analysing a specific trait, to partition the genome along these lines, because it is this conflict—between the cabal and commonwealth—that drives the evolution of the trait value.

Cabals and commonwealths are defined a priori, by partitioning and summing up the coreplicons that, respectively, disfavour and favour the trait distortion under study. The 'individual' is the majority interest within the genome, and so the cabal size can never exceed more than half of the genome, because then it would be the majority ($\theta \leq 0.5$)[53]. In Supplementary Note 8, we calculate some real-world proportional cabal sizes ($\theta$) by dividing the number of genes in a cabal by the total number of genes in a genome. In *Drosophila melanogaster*, a Y chromosome cabal, which favours male biased sex ratio distortion, has a proportional size of $\sim\theta \approx 0.001$[54,55]. In human females, a cabal comprising cytoplasmic elements as well as the X chromosomes, which favours female-biased sex ratio distortion, has a proportional size of $\sim\theta \approx 0.04$[56–58]. In *Escherichia coli*, a cabal made up of horizontally transferrable plasmids, which could favour

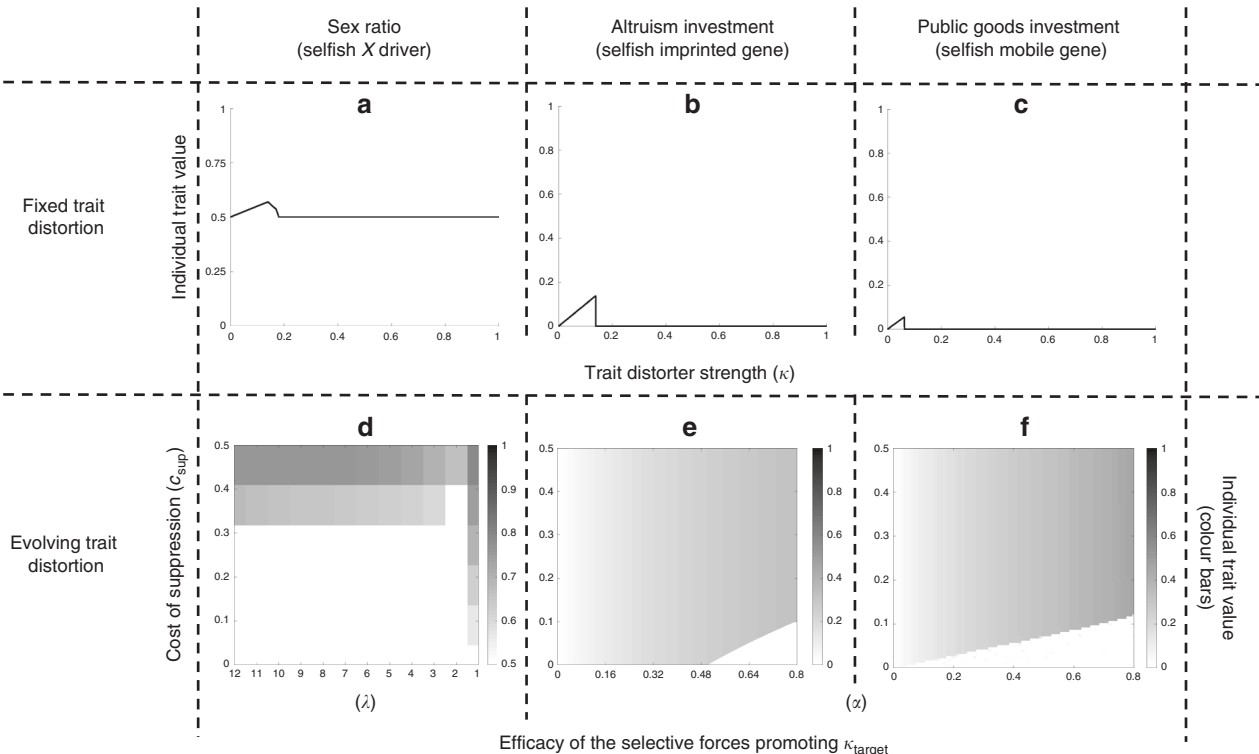

**Fig. 3** Specific biological scenarios. We consider three biological scenarios: **a**, **d** sex ratio distortion by an X driver; **b**, **e** cooperative investment by an imprinted gene (cost and benefit of cooperation are $c = k$ and $b = k^\alpha$, respectively); **c**, **f** cooperative public goods investment by a mobile gene (cost and benefit of cooperation are $c = k$ and $b = 8k^\alpha$, respectively). In all three scenarios, we obtain the same pattern as our illustrative model, that trait distorters have either a minor impact at the individual level or are suppressed ($c_{\mathrm{sup}} = 0.05$; **a**: $\lambda = 2$; **b**, **c**: $\alpha = 0.9$). In **d–f**, we allowed the trait distorters to evolve. We show how the equilibrium level of distortion is determined by the cost of suppression ($c_{\mathrm{sup}}$), and parameters that determine the target level of trait distortion ($k_{\mathrm{target}}$; **d**: $\lambda$; **e**: $\alpha$; **f**: $\alpha$). Trait distortion is low unless there is both a high cost of suppression ($c_{\mathrm{sup}}$) and a high target level of trait distortion (top right of heat maps)

upregulated public goods production[49], varies in size across strains, but has an average of $\sim\theta \approx 0.036$.

For analytical tractability, we start by assuming that new trait distorters and suppressors are introduced at a fixed rate (deterministic). Biologically, new trait distorters and suppressors are likely to arise via some combination of de novo mutation and the acquisition, via gene conversion or transposition, of pre-existing sequences contributing to trait distortion or suppression[35,59,60]. We assume that a trait distorter arises at a new locus within the cabal every $1/(\theta\gamma\rho_{D_1})$ generations, and its dedicated suppressor arises at a locus inside the commonwealth $1/((1-\theta)\gamma\rho_{S_1})$ generations afterwards. $\rho_{D_1}$ and $\rho_{S_1}$, respectively, give the generational per-locus probabilities of generating new trait distorters and suppressors. These probabilities ($\rho_{D_1};\rho_{S_1}$) increase linearly, according to the same gradient, as the baseline mutation rate in the genome, denoted by $\rho$, is increased.

As in our equilibrium models, we assume that unsuppressed trait distorters distort organism traits by the fixed amount $k$, at an individual cost $c_{\mathrm{trait}}(k)$, gaining a meiotic transmission advantage in heterozygotes of $(1 + t(k))/2$. Similarly, we again assume that suppressors are dominant and completely suppress their target trait distorters at the cost $c_{\mathrm{sup}}$, and are facultatively expressed in the presence of their target trait distorter[5–8]. We assume that the trait distortion experienced by an organism is given by the strength of its strongest unsuppressed trait distorter (inter-locus dominance).

We emphasise again that the mechanism by which the trait distorter gains its advantage (meiotic drive) is chosen here purely

for illustrative purposes (see Supplementary Notes 3, 4 and 5 for different mechanisms). We are interested in the subset of selfish genetic elements that gain their selfish benefit by distorting a trait away from the value that maximises individual fitness. The same trait distortion would be favoured across the coreplicon/cabal of which these selfish genetic elements are a part. This contrasts with selfish genetic elements that gain a selfish benefit through their ability to be meiotic drivers, without distorting a trait—such drivers could conceivably arise at any locus in a genome. The key difference here is between meiotic drive (could be favoured at any locus; selfish benefit does not arise via distorting a trait) and selfish genetic elements that gain a benefit by distorting a trait (the specific examples that we consider and model in this paper)[14,15].

We calculate the average and extremes of trait distortion faced by organisms in the population across evolutionary time, for different trait distorter strengths ($k$), and different proportional cabal sizes ($\theta$). Considering trait distorters that do not trigger suppressor invasion ($c_{\mathrm{sup}} > c_{\mathrm{trait}}(k)$), the average trait distortion is trivially given by the strength of the trait distorters available to the cabal ($k$). Considering trait distorters that are suppressed and purged at equilibrium ($c_{\mathrm{sup}} < c_{\mathrm{trait}}(k)$), for analytical tractability, we first consider parameter regimes in which trait distorters are introduced at new loci more slowly than they are purged at old loci, meaning they do not co-segregate.

In 'Long-term trait distortion (exact numerical solution)' in the Methods, we develop a population genetic model based on these assumptions, and solve it numerically to show that individual trait distortion increases and decreases cyclically over evolutionary time, ranging between peaks of $k$ and troughs of 0, as new

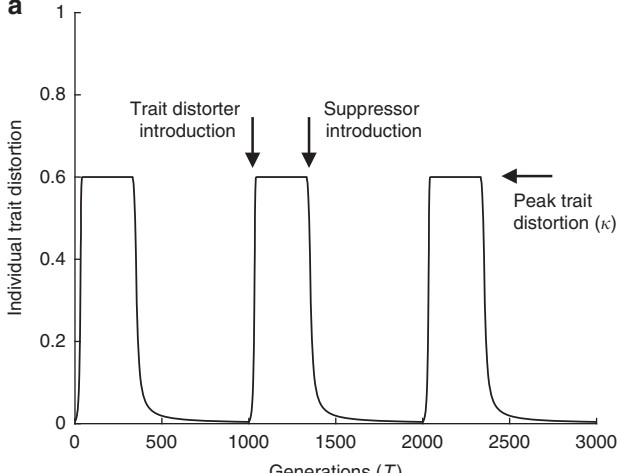

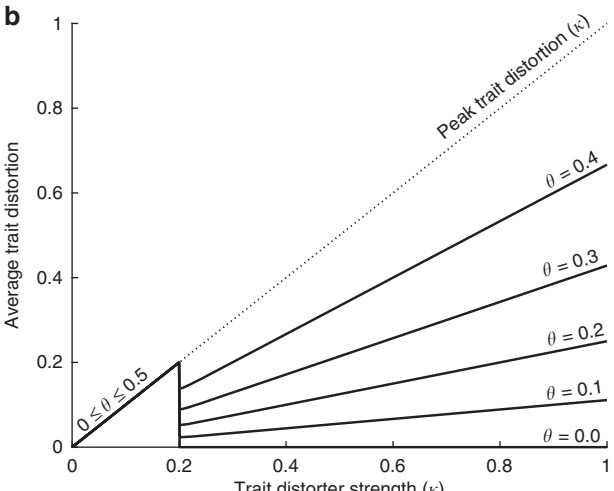

**Fig. 4** The dynamics of conflict when trait distorters do not co-segregate. **a** plots individual trait distortion over evolutionary time, for trait distorters of strength $k = 0.6$. We introduced trait distorters ($D_1$) deterministically at new loci every $1/(\theta\gamma\rho_{D_1})$ generations, and their dedicated suppressors after a lag of $1/((1-\theta)\gamma\rho_{S_1})$ generations. Individual trait distortion increases and decreases cyclically over evolutionary time, between peaks of $k$ and troughs of 0. **b** plots average trait distortion for different proportional cabal sizes ($\theta$), against the strength of trait distorters available to the cabal ($k$). Below a certain threshold strength ($c_{\text{sup}} > c_{\text{trait}}(k)$), suppressors do not invade, and so the resulting trait distortion increases with the strength of the trait distorter ($k$), and is unaffected by proportional cabal size ($\theta$). Above this threshold ($c_{\text{sup}} < c_{\text{trait}}(k)$), suppressors are favoured, and average trait distortion is approximately given by Eq. 6, increasing with trait distorter strength ($k$), although less appreciably for decreased proportional cabal size ($\theta$) (flatter lines) ($c_{\text{sup}} = 0.1$; $t = k$, $c_{\text{trait}} = k/2$, $\rho_{S_1} = 10^{-11}$, $\rho_{D_1} = 10^{-11}$, $\gamma = 10^6$)

trait distorters and suppressors advance and retreat through the population (Fig. 4a). In 'Long-term trait distortion (analytical approximation)' in the Methods, we show that the average trait distortion over these cycles is given by

$$\frac{k\theta\rho_{D_1}}{(1-\theta)\rho_{S_1}}, \tag{6}$$

by making the assumption that the rate of gene frequency equilibration after trait distorter/suppressor introduction is very fast relative to the rate of trait distorter/suppressor introduction

(separation of timescales). For our three specific biological scenarios (Supplementary Notes 3, 4 and 5), the rate of gene frequency equilibration after trait distorter/suppressor introduction varies in each scenario, but these details are inconsequential when the separation of timescales assumption is made, meaning average trait distortion is given by Eq. 6 in each of the three specific biological scenarios. Furthermore, we also found with numerical analysis that Eq. 6 is a good approximation, even when the separation of timescales is relaxed (Fig. 4b).

Smaller proportional cabal sizes ($\theta$) lead to a slower rate of trait distorter introduction relative to suppressor introduction, and so both: (i) an absolute reduction in average trait distortion; and (ii) a reduced effect of distorter strength ($k$) on average trait distortion ($k - \theta$ interaction) (Fig. 4b). In the limit of negligible proportional cabal size ($\theta \to 0$), we recover the result from our Equilibrium models that the proportion of evolutionary time in which a trait distorter is present approaches 0, leading to an average trait distortion of 0 for trait distorters above the threshold of suppression ($c_{\text{sup}} < c_{\text{trait}}(k)$).

Both genome size ($\gamma$) and baseline mutation rate ($\rho$) have no influence on the average trait distortion. Increases in both of these factors leads to a proportional increase in trait distorter introduction rate, and the same proportional increase in suppressor introduction rate, which exactly cancel (Supplementary Note 7, Supplementary Fig. 11).

(6) Trait distortion when trait distorters may co-segregate: We then considered the possibility that different trait distorters may co-segregate for some periods of evolutionary time[59,60]. In 'Agent-based simulation (multiple loci; discrete)' in the Methods, we developed an agent-based simulation that allowed us to investigate the scenario where mutations appear stochastically rather than deterministically. When an individual contains multiple trait distorters, we assume that extent of trait distortion is determined by the strongest trait distorter (inter-locus dominance).

The consequence of allowing trait distorters to co-segregate will depend on mechanistic assumptions about how trait distorters and suppressors act and interact. To capture different ends of the continuum of possibilities, we model two different types of trait distorter, which we term low-sophistication ($D_{1L}$) and high-sophistication ($D_{1H}$) (Supplementary Note 7, Supplementary Fig. 12). High-sophistication trait distorters are only suppressed by dedicated suppressors that evolved to suppress that specific trait distorter, and incur a low cost when inter-locus recessive. In contrast, low-sophistication trait distorters can be suppressed to some extent by any suppressor (background or generalist suppression)[35,59,60], and incur a high cost when inter-locus recessive. High-sophistication trait distorters are more functionally complex, and so are likely to be less mutationally accessible than low-sophistication trait distorters.

We found that, for a sufficiently small proportional cabal size ($\theta \to 0$), trait distorters scarcely co-segregate, and Eq. 6 is recovered. Consequently, for sufficiently small proportional cabal sizes, the average level of trait distortion is again not influenced by genome size ($\gamma$), mutation rate ($\rho$), or the mechanics of trait distorter interaction ($D_{1L}/D_{1H}$).

In contrast, with larger cabals ($\theta \to 0.5$), trait distorters often co-segregate. In this case, the details of genome size ($\gamma$), mutation rate ($\rho$), and trait distorter sophistication ($D_{1L}/D_{1H}$) matter. Specifically, trait distortion may be: (i) greater than Eq. 6 if trait distorters are high sophistication ($D_{1H}$); (ii) lower than Eq. 6 if trait distorters are low sophistication ($D_{1L}$). The deviation from Eq. 6 is exaggerated for increased trait distorter co-segregation, which is promoted by: (i) high genome size ($\gamma$)/mutation rate ($\rho$) (Fig. 5); (iii) low trait distorter strength ($k$), which causes trait distorters to be purged more slowly (Supplementary Note 7,

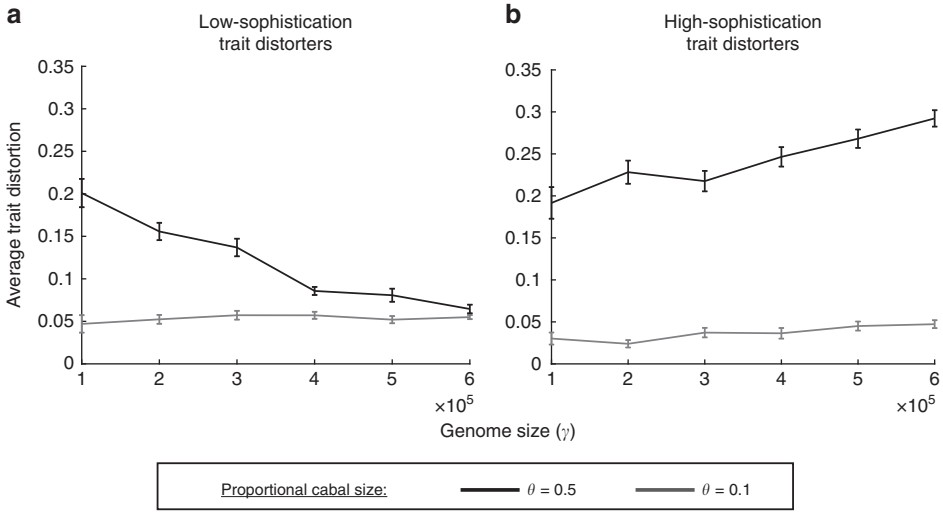

**Fig. 5** Effect of genome size ($\gamma$) on average trait distortion when trait distorters can co-segregate. The average trait distortion is plotted against genome size ($\gamma$), for two different proportional cabal sizes (black: $\theta = 0.5$; grey: $\theta = 0.1$). **a** shows low-sophistication trait distorters ($D_{1L}$), which are partially suppressed by non-dedicated suppressors (background or generalist suppression), and incur a high cost even when inter-locus recessive. **b** shows high sophistication distorters ($D_{1H}$), which are only suppressed by dedicated suppressors, and incur a low cost when inter-locus recessive. The results plotted are an average of 36 runs of our agent-based simulation (error bars represent 1 standard error in each direction), each over $T_{end} = 30,000$ generations, where trait distorters and their dedicated suppressors are introduced stochastically. When proportional cabal size ($\theta$) is smaller, the level of trait distortion is small, and genome size has little influence. When proportional cabal size ($\theta$) is larger, the trait distortion is larger and depends upon genome size. Low-sophistication trait distorters interact counter-productively when co-segregating, meaning trait distortion decreases with genome size, while high-sophistication trait distorters interact productively when co-segregating, meaning trait distortion increases with genome size ($c_{sup} = 0.01$, $t = k$, $c_{trait} =$ Dist/2, $\rho_{S_1} = 4 \times 10^{-9}$, $\rho_{D_{1L}} = 4 \times 10^{-9}$, $\rho_{D_{1H}} = 2 \times 10^{-9}$, $k = 0.5$)

Supplementary Fig. 14); (iv) low trait distorter sophistication ($D_{1L}$), which increases the mutational accessibility of trait distorters. The proportional cabal sizes that make these different factors matter are, however, much larger than we generally find in nature.

(7) Evolution of trait distortion and suppression: We then examined the consequences of allowing the level of trait distortion and suppression to evolve freely at each locus[15]. In 'Agent-based simulation (multiple loci; continuous)' in the Methods, we generalised our agent-based simulation to allow for this, and found that trait distorters evolve increased trait distortion (approaching $k_{target}$) while unsuppressed (Supplementary Note 7, Supplementary Fig. 15). Stronger trait distorters are suppressed and purged more quickly than weaker ones, and are less likely to co-segregate as a result. Consequently, when evolution is permitted at trait distorter loci, average trait distortion again approaches that predicted by Eq. 6, so is less influenced by genome size ($\gamma$), mutation rate ($\rho$), and the mechanics of trait distorter interaction ($D_{1L}/D_{1H}$).

## Discussion

We obtained three main results: First, larger trait distortions are more likely to be suppressed. Consequently, trait distorters will either lead to small trait distortions, with minor fitness consequences, or be suppressed (Figs. 1a and 3a–c). Second, selection on trait distorters favours the evolution of higher levels of trait distortion, which will favour their suppression. Consequently, trait distorters will evolve to bring about their own demise (Figs. 2, 3d–f and 6). Third, if trait distortion is favoured at only a small proportion of the genome (proportionally small cabals), the extent of trait deviation away from the individual level optima is low and unaffected by factors, such as genome size, mutation rate and mechanism of trait distortion (Figs. 4 and 5). The reason for this result is that the influence of all of these factors is determined by proportional cabal size. Overall, these results suggest that even

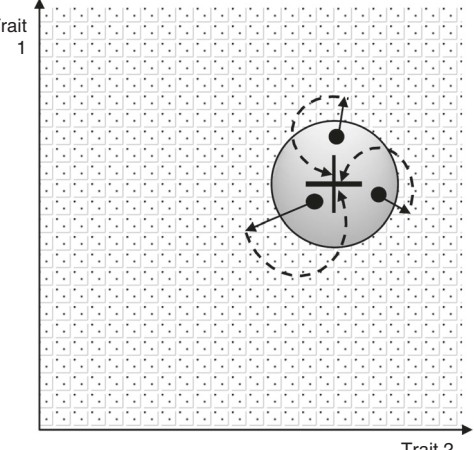

**Fig. 6** Selfish genetic elements evolve to be suppressed by the parliament of genes. The cross represents the position in phenotype space, here defined with respect to two traits, 1 and 2, that maximises the fitness of an individual. The circle surrounding the cross represents the phenotype space where suppression of selfish genetic elements, that have distorted traits 1 or 2, would not be selected for. The surrounding area represents the phenotype space in which the parliament of genes is selected to suppress selfish genetic elements. The three dots represent three possible individuals, each with differently weakly selfish genetic elements, which incur a small fitness cost. Because these deviations from individual fitness maximisation are only slight, costly suppression of the weakly selfish genetic elements does not evolve. However, the selfish genetic elements will evolve to become more distorting (solid arrows), bringing individuals into the area of phenotype space where they will be suppressed and individual fitness maximisation (the black cross) is regained (dashed arrows)

if there is substantial potential for genetic conflict, trait distorters will have relatively little influence at the individual level, in support of Leigh's[28] parliament of genes hypothesis.

Suppressing trait distorters: We have shown that suppressors spread when the cost of suppression is lower than the fitness cost imposed by trait distortion ($c_{trait}(k) > c_{sup}$). The individual fitness cost of pre-translational suppression at a single locus is likely to be low. For example, a molecularly characterised suppressor (*nmy*) destroys the messenger RNA transcripts of a sex ratio distorter (*Dox*) via RNA interference (RNAi), the costs of which are likely to be negligible at the individual level[46,47,60,61]. Consequently, in order to not be suppressed, a trait distorter would have to have relatively negligible influence on a trait, or influence a trait that has a negligible influence on fitness. Furthermore, we also showed that selection on trait distorters will often favour higher level trait distortion, bringing trait distorters into the region where $c_{trait}(k) > c_{sup}$, and hence where suppression is favoured (Figs. 2, 3 and 6).

Our analyses have focused on selfish genetic elements that increase their own transmission by manipulating some organism trait in a specific direction[15,17]. Examples include the sex ratio distorters and public goods genes considered in our specific models. We focused on such 'trait distorters' because they can have substantial influences on the traits of organisms, even when at fixation. In contrast, we have not considered selfish genetic elements, such as transposons and meiotic drivers, that do not need to manipulate organism traits in order to give themselves a selfish propagation advantage[43]. We have not considered such selfish genetic elements because: (i) they do not distort traits away from individual maxima; and (ii) the cost of such drivers makes them disfavoured across the entire genome, leading to selection to attenuate that cost.

Our Dynamics models have validated various verbal arguments that have previously been made for the parliament of genes hypothesis. We found that, if trait distortion is only favoured across a small proportion of the genome (proportionally small cabal), the trait distortion experienced by individuals is likely to be low, and unaffected by details such as genome size, mutation rate and mechanism of trait distortion. Empirically, cabals typically comprise small proportions of genomes[54,56]. Furthermore, more sophisticated trait distorters, with the potential to interact synergistically with each other, are likely to have a lower mutational accessibility, and so are more likely to be suppressed and purged before they have a chance to co-segregate. Real-world examples of trait distortion are typically caused by lone genes, or genes that do not interact synergistically[14,60]. In contrast, complex adaptations are typically underpinned by multitudes of synergistically interacting genes residing in the parliamentary majority (commonwealth)[23].

We are not claiming that appreciable trait distortion will never evolve, or that biological details will never matter[14,32,59,60]. Instead, our results suggest that the modal outcome will be a relative lack of trait distortion. This conclusion is supported empirically by cases where appreciable distortion is only revealed in hybrid crosses, implying that trait distorters are generally suppressed[62]. Furthermore, we find that, after suppression has evolved, trait distorters are generally purged from the population at equilibrium. If suppressors are constitutively expressed (obligate), trait distorters are not purged from the population, but in these cases, suppressors spread to fixation (Supplementary Note 6). Regardless of the extent to which suppressors are constitutive, there is negligible polymorphism in at least one locus, meaning trait distortion is unlikely to be revealed by mating within a population[38]. When trait distorters are not purged from the population, trait distortion will be revealed by matings between populations/species[62].

Sex ratio distorters as a case study: The relatively large literature on sex ratio distorters offers a chance for us to assess the validity of our models, and their predictions. In Supplementary Note 3, we detail how our assumptions are consistent with the biology of sex ratio distorters and their suppressors. For example, X drivers increase their own transmission by killing Y bearing sperm, and hence producing a female-biased offspring sex ratio. This comes at a cost to the rest of the genome through both a reduction in sperm number, and through Fisherian selection disfavouring the more common sex (females). The scope of the parliament of genes to act against such drivers is shown by the fact that, in most species in which an X driver is present, suppressors have been found on both the autosomes and the Y chromosome[36]. Our assumptions about how suppressors act, and the cost of suppression, are analogous to those in a molecularly characterised suppressor (*Nmy*) of a sex ratio distorter (*Dox*)[46,60,61]; and more generally to suppressors that act pre-translationally[63,64].

Our model predictions are consistent with the available data on X drivers in *Drosophila*. As predicted by our model: (1) Across natural populations of *Drosophila simulans*, there is a positive correlation between the extent of sex ratio distortion and the extent of suppression[65]. (2) In both *Drosophila mediopunctata* and *D. simulans* the presence of an X-linked driver led to the experimental evolution of suppression[66,67]. In addition, consistent with our model: (3) In natural populations of *D. simulans*, the prevalence of an X driver has been shown to sometimes decrease under complete suppression[68]. (4) Crossing different species of *Drosophila* has been shown to lead to appreciable sex ratio deviation, by unlinking trait distorters from their suppressors, and hence revealing previously hidden trait distorters[62]. Work on other sex ratio distorters has also shown that suppressors can spread extremely quickly from rarity, reaching fixation in as little as ~5 generations[69].

Individual fitness maximisation: We emphasise that when the assumption of individual fitness maximisation is made in behavioural and evolutionary ecology, it is not being assumed that natural selection produces perfect fitness maximisers[5]. Many factors could constrain adaptation, such as genetic architecture, mutation and phylogenetic constraints[70,71]. Instead, the assumption of fitness maximisation is used as a basis to investigate the selective forces that have favoured particular traits (adaptations). The aim is not to test if organisms maximise fitness, or behave 'optimally', but rather to try to understand the selective forces favouring particular traits or behaviours[2]. We have examined how the parliament of genes prevents selfish genetic elements from constraining adaptation, focusing on the maintenance, rather than the emergence, of traits (Supplementary Discussion).

To conclude, debate over the validity of assuming individual level fitness maximisation has usually revolved around whether selfish genetic elements are common or rare[4,20,21,24,72]. We have shown that that even if selfish genetic elements are common, they will tend to be either weak and negligible, or suppressed. This suggests that even if there is the potential for appreciable genetic conflict, individual level fitness maximisation will still often be a reasonable assumption. This allows us to explain why certain traits, especially the sex ratio, have been able to provide such clear support for both individual level fitness maximisation and genetic conflict[9].

## Methods

**Trait distorter population frequency**. We ask when a rare trait distorter ($D_1$) can invade a population fixed for the trait non-distorter ($D_0$). We take Eq. 1, set $p' = p = p^*$, and solve to find two possible equilibria: $p^* = 0$ (trait non-distorter fixation) and $p^* = 1$ (trait distorter fixation). The trait distorter ($D_1$) can invade from rarity when the $p^* = 0$ equilibrium is unstable, which occurs when the differential of $p'$

with respect to $p$, at $p^\star = 0$, is >1. The trait distorter invasion criterion is therefore $c_{trait}(k) < t(k)(1 - c_{trait}(k))$.

We now ask what frequency the trait distorter ($D_1$) will reach after invasion. The trait distorter ($D_1$) can spread to fixation if the $p^\star = 1$ equilibrium is stable, which requires that the differential of $p'$ with respect to $p$, at $p^\star = 1$, is <1. This requirement always holds true, demonstrating that there is no negative frequency dependence on the trait distorter, and that it will always spread to fixation after its initial invasion.

**Suppressor invasion condition.** We ask when the suppressor ($S_1$) can spread from rarity in a population in which the trait distorter ($D_1$) and non-suppressor ($S_0$) are fixed at equilibrium. We derive the Jacobian stability matrix for this equilibrium, which is a matrix of each genotype frequency ($x_{00}'$, $x_{01}'$, $x_{10}'$, $x_{11}'$) differentiated by each genotype frequency in the prior generation ($x_{00}$, $x_{01}$, $x_{10}$, $x_{11}$), at the equilibrium position given by $x_{00}^\star = 0$, $x_{01}^\star = 0$, $x_{10}^\star = 1$, $x_{11}^\star = 0$:

$$J = \begin{pmatrix} 1-t & \frac{1-c_{sup}}{2(1-c_{trait})} & 0 & 0 \\ 0 & \frac{1-c_{sup}}{2(1-c_{trait})} & 0 & 0 \\ t-1 & \frac{-3(1-c_{sup})}{2(1-c_{trait})} & 0 & \frac{-(1-c_{sup})}{1-c_{trait}} \\ 0 & \frac{1-c_{sup}}{2(1-c_{trait})} & 0 & \frac{1-c_{sup}}{1-c_{trait}} \end{pmatrix}, \quad (7)$$

The suppressor can invade when the equilibrium is unstable, which occurs when the leading eigenvalue is greater than one. The leading eigenvalue is $(1 - c_{sup})/(1 - c_{trait})$, meaning the suppressor invasion criterion is $c_{trait} > c_{sup}$.

**Equilibrium trait distorter and suppressor frequencies.** We ask what frequency the trait distorter ($D_1$) and suppressor ($S_1$) will reach after initial suppressor ($S_1$) invasion. We assume that the suppressor is introduced from rarity when the trait distorter has reached the population frequency given by $f$ ($x_{00} \to f$, $x_{10} \to 1 - f$, $\{x_{01}, x_{11}\} \to 0$). We numerically iterate Eqs. 2–5, over successive generations, until equilibrium has been reached. At equilibrium, for all parameter combinations ($f$, $t$, $c_{sup}$, $c_{trait}$), the suppressor reaches an internal equilibrium and the trait distorter is lost from the population ($x_{00}^\star + x_{01}^\star = 1$, $x_{10}^\star = 0$, $x_{11}^\star = 0$). This equilibrium arises because trait distorter presence gives the suppressor ($S_1$) a selective advantage, leading to high suppressor frequency, which in turn reverses the selective advantage of the trait distorter ($D_1$), leading to trait distorter loss and suppressor equilibration.

**Non-equilibrium trait distortion.** We consider a trait distorter that is suppressed and therefore purged at equilibrium ($c_{trait} > c_{sup}$), and ask to what extent it can contribute to individual trait distortion in the period after its initial invasion, but before its eventual loss (non-equilibrium). We introduce the trait distorter ($D_1$) and suppressor ($S_1$) from rarity and numerically iterate our recursions until the trait distorter has been purged from the population (or a cap of 20,000,000 generations has been reached). We vary parameters between $0 \le t \le 1$, $c_{sup} < c_{trait} \le 1$, $0 \le c_{sup} \le 1$.

We find that a higher cost of trait distortion ($c_{trait}$) relative to suppression ($c_{sup}$) leads to shorter non-equilibrium maintenance of the trait distorter in the population. This is because the cost of trait distortion relative to suppression mediates selection on the suppressor (Methods: 'Suppressor invasion condition'). We find that a higher transmission bias ($t$) leads to longer non-equilibrium maintenance of the trait distorter in the population, but this effect is diluted as the cost of trait distortion ($c_{trait}$) is increased relative to suppression ($c_{sup}$) (Supplementary Note 2, Supplementary Fig. 1). Stronger trait distorters (with higher $k$, leading to higher $c_{trait}$ and $t$) are therefore generally suppressed and purged more rapidly than weaker trait distorters (Fig. 1b). Exceptions are trait distorters that reduce individual fitness relatively negligibly after the point ($k$) at which suppression is favoured, such that $\frac{dt}{dk} / \frac{dc_{trait}}{dk}$ is very high for values of $k$ satisfying $c_{sup} < c_{trait}(k)$.

**Invasion of a mutant trait distorter.** We ask when a mutant trait distorter ($D_2$) will invade against a resident trait distorter ($D_1$) that is unsuppressed and at fixation ($k \ne \hat{k}$). We write recursions detailing the generational frequency changes in the six possible gametes, $D_0/S_0$, $D_0/S_1$, $D_1/S_0$, $D_1/S_1$, $D_2/S_0$, $D_2/S_1$, with current generation frequencies denoted, respectively by $x_{00}$, $x_{01}$, $x_{10}$, $x_{11}$, $x_{20}$, $x_{21}$, and next-generation frequencies denoted with an appended dash ('):

$$\bar{w}x_{00}' = x_{00}x_{00} + x_{00}x_{01} + (1 - t(k))(1 - c_{trait}(k))x_{00}x_{10} \\ + \left(\left(1 - c_{sup}\right)/2\right)x_{00}x_{11} + (1 - t(k))(1 - c_{trait}(\hat{k})) \\ x_{00}x_{20} + \left(\left(1 - c_{sup}\right)/2\right)x_{00}x_{21} + \left(\left(1 - c_{sup}\right)/2\right) \\ x_{01}x_{10} + \left(\left(1 - c_{sup}\right)/2\right)x_{01}x_{20}, \quad (8)$$

$$\bar{w}x_{01}' = x_{00}x_{01} + \left(\left(1 - c_{sup}\right)/2\right)x_{00}x_{11} + \left(\left(1 - c_{sup}\right)/2\right) \\ x_{00}x_{21} + x_{01}x_{01} + \left(\left(1 - c_{sup}\right)/2\right)x_{01}x_{10} + \left(1 - c_{sup}\right)x_{01}x_{11} \\ + \left(\left(1 - c_{sup}\right)/2\right)x_{01}x_{20} + \left(1 - c_{sup}\right)x_{01}x_{21}, \quad (9)$$

$$\bar{w}x_{10}' = (1 + t(k))(1 - c_{trait}(k))x_{00}x_{10} + \left(\left(1 - c_{sup}\right)/2\right) \\ x_{00}x_{11} + \left(\left(1 - c_{sup}\right)/2\right)x_{01}x_{10} + (1 - c_{trait}(k)) \\ x_{10}x_{10} + \left(1 - c_{sup}\right)x_{10}x_{11} + (1 + t(k) - t(\hat{k})) \\ (1 - c_{trait}(\max(k, \hat{k})))x_{10}x_{20} + \left(\left(1 - c_{sup}\right)/2\right)x_{10}x_{21} \\ + \left(\left(1 - c_{sup}\right)/2\right)x_{11}x_{20}, \quad (10)$$

$$\bar{w}x_{11}' = \left(\left(1 - c_{sup}\right)/2\right)x_{00} \times_{11} + \left(\left(1 - c_{sup}\right)/2\right) \\ x_{01}x_{10} + \left(1 - c_{sup}\right)x_{01}x_{11} + \left(1 - c_{sup}\right) \\ x_{10}x_{11} + \left(\left(1 - c_{sup}\right)/2\right)x_{10}x_{21} \\ + \left(1 - c_{sup}\right)x_{11}x_{11} + \left(\left(1 - c_{sup}\right)/2\right) \\ x_{11}x_{20} + \left(1 - c_{sup}\right)x_{11}x_{21}, \quad (11)$$

$$\bar{w}x_{20}' = (1 + t(\hat{k}))(1 - c_{trait}(\hat{k}))x_{00}x_{20} + \left(\left(1 - c_{sup}\right)/2\right) \\ x_{00}x_{21} + \left(\left(1 - c_{sup}\right)/2\right)x_{01}x_{20} + (1 - t(k) + t(\hat{k})) \\ (1 - c_{trait}(\max(k, \hat{k})))x_{10}x_{20} + \left(\left(1 - c_{sup}\right)/2\right) \\ x_{10}x_{21} + \left(\left(1 - c_{sup}\right)/2\right)x_{11}x_{20} + (1 - c_{trait}(\hat{k})) \\ x_{20}x_{20} + \left(1 - c_{sup}\right)x_{20}x_{21}, \quad (12)$$

$$\bar{w}x_{21}' = \left(\left(1 - c_{sup}\right)/2\right)x_{00}x_{21} + \left(\left(1 - c_{sup}\right)/2\right) \\ x_{01}x_{20} + \left(1 - c_{sup}\right)x_{01}x_{21} + \left(\left(1 - c_{sup}\right)/2\right)x_{10}x_{21} + \left(\left(1 - c_{sup}\right)/2\right) \\ x_{11}x_{20} + \left(1 - c_{sup}\right)x_{11}x_{21} + \left(1 - c_{sup}\right)x_{20}x_{21} + \left(1 - c_{sup}\right)x_{21}x_{21}, \quad (13)$$

where $\bar{w}$ is the average fitness of individuals in the current generation, and equals the sum of the right-hand side of the system of equations. The mutant trait distorter can invade when the equilibrium given by $x_{00}^\star = 0$, $x_{01}^\star = 0$, $x_{10}^\star = 1$, $x_{11}^\star = 0$, $x_{20}^\star = 0$, $x_{21}^\star = 0$ is unstable, which occurs when the leading eigenvalue of the Jacobian stability matrix for this equilibrium is >1. Testing for stability in this way, we find that, if the mutant trait distorter is weaker than the resident, it can never invade. If the mutant trait distorter is stronger than the resident, it invades from rarity when $\Delta t(1 - c_{trait}(\hat{k})) > \Delta c_{trait}$, where $\Delta t = t(\hat{k}) - t(k)$, $\Delta c_{trait} = c_{trait}(\hat{k}) - c_{trait}(k)$.

The implication is that, if trait distortion is initially low, and mutant trait distorters are successively introduced, each deviating only very slightly from the resident trait distorter from which they are derived, such that $\hat{k} = k + \delta$, where $\delta$ is very small ('δ-weak selection'[48]), then trait distorters will approach a 'target' strength at which $\frac{dt}{dk}(1 - c_{trait}) = \frac{dc_{trait}}{dk}$. In the absence of suppression, this target ($k_{target}$) is the equilibrium level of trait distortion ($k^\star = k_{target}$). However, if mutant trait distorters ($D_2$) are allowed to deviate appreciably from residents ($D_1$) (strong selection), then trait distorters may invade even if they overshoot the target ($\hat{k} > k_{target}$). In the absence of suppression, $k_{target}$ is then not the equilibrium level of trait distortion, but rather, the minimum equilibrium level of trait distortion ($k^\star \ge k_{target}$) (Supplementary Note 2, Supplementary Fig. 2b).

We could alternatively have assumed that an individual's trait is distorted according to the average strength of its alleles (additive gene interactions), rather than according to the stronger (higher $k$) allele (dominance). Such an assumption leads to a single invasion criterion for a mutant trait distorter, regardless of whether the mutant trait distorter is stronger or weaker than the resident trait distorter, given by: $\Delta t(2 - c_{trait}(k) - c_{trait}(\hat{k})) > \Delta c_{trait}$. In the absence of suppression, this leads to an equilibrium level of trait distortion ($k^\star$), which holds even under strong selection, and satisfies $2\frac{dt}{dk}(1 - c_{trait}) = \frac{dc_{trait}}{dk}$.

**Equilibrium allele frequencies after mutant invasion.** We ask what equilibrium state will arise after the invasion of a mutant trait distorter. We assume that the mutant trait distorter ($D_2$) is introduced from rarity when the resident trait distorter ($D_1$) has reached the population frequency given by $q$. We numerically iterate Eq. 8–13, over successive generations, until equilibrium has been reached. At equilibrium, for all parameter combinations ($q$, $t(k)$, $t(\hat{k})$, $c_{sup}$, $c_{trait}(k)$, $c_{trait}(\hat{k})$), the resident trait distorter ($D_1$) is lost from the population ($x_{10}, x_{11} = 0$), with either the mutant trait distorter ($D_2$) and non-suppressor ($S_0$) at fixation ($x_{20}^\star = 1$), or the trait non-distorter ($D_0$) at fixation alongside the suppressor ($S_1$) at an internal equilibrium ($x_{00}^\star + x_{01}^\star = 1$). The latter scenario arises if the mutant trait distorter triggers suppressor invasion ($c_{sup} < c_{trait}(\hat{k})$). This equilibrium arises because mutant trait distorter presence gives the suppressor ($S_1$) a selective advantage, leading to high

suppressor frequency, which in turn reverses the selective advantage of trait distortion, leading to trait distorter ($D_1$, $D_2$) loss and suppressor equilibration.

**Agent-based simulation (single trait distorter locus).** We construct an agent-based simulation to ask what level of trait distortion evolves when continuous variation is permitted at trait distorter and suppressor loci. We model a population of $N = 2000$ individuals and track evolution at two autosomal loci: a trait distorter locus and a suppressor locus. Each individual has two alleles at the trait distorter locus, with strengths denoted by $k_a$ and $k_b$, and two alleles at the suppressor locus, with strengths denoted by $m_a$ and $m_b$ (diploid). Strengths can take any continuous value between 0 and 1. We assume that, for both loci, the strongest (highest value) allele within an individual is dominant. The absolute fitness of an individual with at least one active meiotic driver ($\max(k_a, k_b) > 0$) is: $1 - c_{\text{trait}}(\max(k_a, k_b))(1 - \max(m_a, m_b)) - c_{\text{sup}}\max(m_a, m_b)$, and the absolute fitness of an individual lacking an active trait distorter ($\max(k_a, k_b) = 0$) is 1. The function $c_{\text{trait}}(\max(k_a, k_b))$ is given an explicit form in simulations (Supplementary Note 2, Supplementary Fig. 2).

In each generation, there are $N$ breeding pairs. To fill each position in each breeding pair, individuals are drawn from the population, with replacement, with probabilities given by their fitness (hermaphrodites). Breeding pairs then reproduce to produce one offspring, before dying (non-overlapping generations). Alleles at the suppressor locus are inherited in Mendelian fashion. Alleles at the trait distorter locus may drive, meaning the parental allele of strength $k_a$ is inherited, rather than the allele of strength $k_b$, with the probability $(1 + (t(k_a) - t(k_b))(1 - \max(m_a, m_b)))/2$. The transmission bias function, $t$, is given an explicit form in simulations (Supplementary Note 2, Supplementary Fig. 2). Each generation, trait distorter and suppressor alleles have a 0.01 chance of mutating to a new value, which is drawn from a normal distribution centred around the pre-mutation value, with variance 0.2, and truncated between 0 and 1. We track the population average trait distorter strength, denoted by $E[k]$, and suppressor strength, denoted by $E[m]$, over 20,000 generations. We see that, allowing for continuous variation at the trait distorter and suppressor loci, if the cost of suppression ($c_{\text{sup}}$) is not excessively high, trait distortion at equilibrium is either low or nothing (Fig. 2a; Supplementary Note 2, Supplementary Fig. 2b).

**Long-term trait distortion (exact numerical solution).** We ask how the trait distortion experienced by organisms changes across evolutionary time as new trait distorters and suppressors are continuously introduced and lost from a population. We construct a population genetic model and solve it numerically and exactly. We introduce a trait distorter from rarity and iterate our recursion for an unsuppressed trait distorter (Eq. 1) from $T = 1$ to $T = 1/((1 - \theta)\gamma\rho_{S_1})$ generations. During this period, the trait distortion experienced by individuals rises to a peak of $k$, corresponding to the strength of trait distorters available to the cabal. We then introduce a suppressor from rarity and iterate our recursions for trait distorter-suppressor co-segregation (Eqs. 2–5), from $T = 1/((1 - \theta)\gamma\rho_{S_1})$ until the trait distorter has been purged ($T = X$). During this period, the trait distortion experienced by individuals falls to a trough of 0.

Average trait distortion over evolutionary time is given by weighting average trait distortion during the interval $T = \{1, 2, ..., X\}$ by the proportion of evolutionary time in which a trait distorter is segregating in the population ($X(\theta\gamma\rho_{D_1})$). This methodology provides exact, numerical values for average trait distortion. These values correspond closely to the analytical approximation for average trait distortion (Eq. 6), which is derived under a separation of timescales assumption (Methods: 'Long-term trait distortion (analytical approximation)'; Fig. 4).

**Long-term trait distortion (analytical approximation).** When a trait distorter is initially introduced into the population, it will spread, and the population will equilibrate when the trait distorter reaches fixation (Methods: 'Long-term trait distortion (exact numerical solution)'). Similarly, when a suppressor is initially introduced into the population, it will spread if its target trait distorter is sufficiently costly ($c_{\text{sup}} < c_{\text{trait}}(k)$), and the population will equilibrate when the suppressor's target trait distorter is purged from the population (Methods: 'Long-term trait distortion (exact numerical solution)'). We assume that, after the introduction of a new trait distorter or suppressor, the rate at which gene frequencies equilibrate is very fast relative to the rate at which new trait distorters and suppressors are introduced at new loci (separation of timescales).

On this assumption, we can partition evolutionary time into two repeating periods. In the first period, comprising the $1/((1 - \theta)\gamma\rho_{S_1})$ generations in between trait distorter and suppressor introduction, individual trait distortion is $k$. In the second period, comprising the following $1/(\theta\gamma\rho_{D_1}) - 1/((1 - \theta)\gamma\rho_{S_1})$ generations, and ending when the next trait distorter is introduced at a new locus, individual trait distortion is 0. We average over these two time periods to calculate the average trait distortion experienced by individuals across evolutionary time (Eq. 6).

**Agent-based simulation (multiple loci; discrete).** We build on the agent-based model detailed in Methods: 'Agent-based simulation (single trait distorter locus)' to capture the evolutionary dynamics of arbitrarily large numbers of co-segregating trait distorters and suppressors across the genome. The specific details of how mate partners are attributed (e.g. panmictic; hermaphrodite), and how the population is

sampled to implement fitness effects (e.g. non-overlapping generations), are fully described in Methods: 'Agent-based simulation (single trait distorter locus)'. We model a diploid population of $N = 2000$ individuals, each with $\gamma = 10^6$ loci, $\theta\gamma$ of which constituting the cabal and $(1 - \theta)\gamma$ of which constituting the commonwealth.

We assume that each locus across the genome is initially 'dormant'. The alleles segregating in the population at dormant loci are neutral with respect to trait distortion and suppression. Loci are activated when the alleles segregating there have drifted to lie one mutational step away from distortion or suppression. For a given dormant locus in the cabal and in the commonwealth, the generational activation probability is given, respectively, by $\rho_{D_1}$ and $\rho_{S_1}$. Each successively activated cabal and commonwealth locus is indexed with a consecutive integer within the respective sets $I_{\text{cabal}} = \{1, 2, ..., n_{\text{cabal}}\}$ and $I_{\text{commonwealth}} = \{1, 2, ..., n_{\text{commonwealth}}\}$, where $n_{\text{cabal}}$ and $n_{\text{commonwealth}}$ give respectively the total number of activated cabal and commonwealth loci, which increase as generations ($T$) pass. After locus activation, alleles mutate between functional and neutral forms with a generational probability of 0.001. If, at any time, all trait distorters ($i \in I_{\text{cabal}}$) have dedicated suppressors ($i \in I_{\text{commonwealth}}$), such that $n_{\text{cabal}} = n_{\text{commonwealth}}$, further commonwealth loci cannot be activated until new trait distorters arise ($n_{\text{cabal}} > n_{\text{commonwealth}}$). If trait distorters are low-sophistication as opposed to high-sophistication, the generational cabal locus activation probability ($\rho_{D_1}$) is increased by a factor two (such that $\rho_{D_{1L}} = 2 {}^*\rho_{D_{1H}}$).

For each individual, the set $I_{\text{distorter}} \subseteq I_{\text{cabal}}$ comprises every locus within the cabal where one (heterozygous) or two (homozygous) trait distorters are present. A given suppressor at a locus within the commonwealth ($i \in I_{\text{commonwealth}}$) is only expressed if its target trait distorter ($i \in I_{\text{distorter}}$) is also present in the individual. However, if expressed, a given suppressor ($i \in I_{\text{commonwealth}}$) may also contribute to the 'background' suppression of unsuppressed non-target trait distorters ($I_{\text{distorter}}\backslash i$), at a fraction $z$ of its usual strength. We assume that, for low-sophistication trait distorters ($D_{1L}$), $z = 0.5$, and for high-sophistication trait distorters ($D_{1H}$), $z = 0$.

The total suppression faced by a trait distorter ($i \in I_{\text{distorter}}$) is therefore $\text{TotSup}_i = 1$ if its dedicated suppressor is present in the individual, or $\text{TotSup}_i = \min(zq, 1)$ if its dedicated suppressor is absent, where $q$ is the number of expressed suppressors present in the individual, and where the 'min' notation indicates that the total suppression cannot exceed 1 (complete suppression). The total cost of suppression for an individual is $c_{\text{sup}} \sum_{i \in I_{\text{distorter}}} \text{TotSup}_i$. The least suppressed trait distorter in each individual ($i_{\text{dom}} \in I_{\text{distorter}}$) exerts inter-locus dominance, and causes a trait distortion of $\text{Dist} = \max_{i \in I_{\text{distorter}}} ((1 - \text{TotSup}_i)k)$. The individual cost of trait distortion, which is given by $c_{\text{trait}}(\text{Dist})$, increases monotonically with the extent that the trait is distorted $\left(\frac{dc_{\text{trait}}}{d\text{Dist}} \geq 0\right)$.

Expression of the remaining 'inter-locus recessive' trait distorters ($I_{\text{distorter}}\backslash i_{\text{dom}}$) leads to a pool of gene products with an abundance that is proportional to: $\text{Waste} = \sum_{\substack{i \in I_{\text{distorter}} \\ i \neq i_{\text{dom}}}} ((1 - \text{TotSup}_i)k)$. The individual cost arising from inter-locus recessive trait distorters, which is given by $c_{\text{rec}}$, increases monotonically with the size of the pool of redundant gene products $\left(\frac{dc_{\text{rec}}}{d\text{Waste}} \geq 0\right)$. We assume that, for low-sophistication trait distorters ($D_{1L}$), the individual cost arising from any one inter-locus recessive trait distorter is equal to the cost of trait distortion itself $\left(c_{\text{trait}}(\text{Dist}) = \frac{c_{\text{rec}}(\text{Waste})}{|I_{\text{distorter}}| - 1} \geq 0\right)$. For high-sophistication trait distorters ($D_{1H}$), this cost is lower relative to the cost of trait distortion $\left(c_{\text{trait}}(\text{Dist}) = \frac{5(c_{\text{rec}}(\text{Waste}))}{3(|I_{\text{distorter}}| - 1)} \geq 0\right)$. The total fitness (viability) of an individual is then given by: $1 - c_{\text{trait}}(\text{Dist}) - c_{\text{rec}}(\text{Waste}) - c_{\text{sup}} \sum_{i \in I_{\text{distorter}}} \text{TotSup}_i$.

We define the set $I_{\text{het}} \subseteq I_{\text{distorter}} \subseteq I_{\text{cabal}}$ as the collection of loci in an individual at which one (heterozygous) trait distorter, as opposed to two (homozygous) trait distorters, are present. The trait distorters at these loci ($I_{\text{het}}$) drive at meiosis, as a unit. The least suppressed trait distorter in the group pulls the unit through meiosis, meaning the group of trait distorters (at loci $I_{\text{het}}$) is inherited by each offspring with the probability $(1 + \max_{i \in I_{\text{het}}} (1 - \text{TotSup}_i)k)/2$.

**Agent-based simulation (multiple loci; continuous).** We adapt the simulation model detailed in Methods: 'Agent-based simulation (multiple loci; discrete)' so that trait distorters and suppressors are not of fixed strength (of $k$ and 1, respectively), but are free to evolve continuously between 0 and 1. Homologous alleles at activated cabal loci ($i \in I_{\text{cabal}}$) have strengths $k_{ai}$ and $k_{bi}$, and homologous alleles at activated commonwealth loci ($i \in I_{\text{commonwealth}}$) have strengths $m_{ai}$ and $m_{bi}$. Within an individual, the loci bearing trait distorters ($I_{\text{distorter}} \subseteq I_{\text{cabal}}$) each satisfy $\max(k_{ai}, k_{bi}) > 0$. Each trait distorter (at locus $i \in I_{\text{distorter}}$) is suppressed to the following extent:

$$\text{TotSup}_i = \min\left(\max(m_{ai}, m_{bi}) + z \sum_{\substack{j \in I_{\text{distorter}} \\ j \neq i}} \max(m_{aj}, m_{bj}), 1\right).$$

Within an individual, the strongest trait distorter (after suppression) is inter-locus dominant ($i_{\text{dom}} \in I_{\text{distorter}}$), and distorts the individual trait

by: $\text{Dist} = \max_{i \in I_{\text{distorter}}} \big((1 - \text{TotSup}_i)\max(k_{ai}, k_{bi})\big)$. The inter-locus recessive trait distorters ($I_{\text{distorter}} \backslash i_{\text{dom}}$) bring about an additional individual level cost of $c_{\text{rec}}(\text{Waste})$, which is a monotonically increasing function of

$$\text{Waste} = \sum_{\substack{i \in I_{\text{distorter}} \\ i \neq i_{\text{dom}}}} \big((1 - \text{TotSup}_i)\max(k_{ai}, k_{bi})\big).$$

If an allele is more trait-distorting than its homologue ($k_{ai}$ vs. $k_{bi}$), it can drive at meiosis. The strongest alleles across each homologous pair drive together as a single unit. The unit is inherited by each offspring with the probability $\left(1 + \max_{i \in I_{\text{distorter}}}(1 - \text{TotSup}_i)\text{abs}(k_{ai} - k_{bi})\right)/2$. Every generation, each allele at an activated locus has a 0.01 chance of mutating to a new strength, which is drawn from a normal distribution centred around the pre-mutation strength, with variance 0.2, and truncated between 0 and 1.

**Reporting summary**. Further information on research design is available in the Nature Research Reporting Summary linked to this article.

## Data availability
The data that support the findings of this study are available upon request.

## Code availability
Simulation code is available upon request.

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

## Acknowledgements

We thank Geoff Wild, Alan Grafen, Egbert Giles Leigh, Jr., Guy Cooper, Sam Levin, Asher Leeks, Matishalin Patel and Tom Hitchcock for comments on the manuscript; Anna Dewar for sharing data on the ratio of plasmid to chromosomal genes in *E. coli*.

## Author contributions

T.W.S. and S.A.W. designed the study and wrote the paper. T.W.S. carried out mathematical analysis.

## Competing interests

The authors declare no competing interests.
