## [Peer Review File · Nature Communications]

Reviewers' Comments:

Reviewer #1:

Remarks to the Author:

GENERAL COMMENTS

In this paper Scott and West examines the apparent contradiction between the assumption, common among behavioral ecologists, that organisms should appear designed to maximize their (inclusive) fitness and the presence of selfish genetic elements that undermines the organismal 'unity-of-purpose' required for this. The authors develop a series of models of distorters and suppressors to argue that selfish genetic elements will either have too weak fitness costs to be important, or, if the fitness costs to the individual organism are large, they will be readily suppressed.

There is a lot to like about this paper. The contradiction highlighted is an important and understudied one, the text is clearly written and the argument is easy to follow, and (to the best of my ability) the math seems correct. I also appreciated that they adapted their models to three different biological examples. Yet, I am not convinced that the authors actually provide any new answers the stated contradiction.

To start, the theoretical justification for the design maximization argument is usually found in optimization theory, as established by Alan Grafen and endorsed by the authors. While there are many critics of this approach, I am not one of them. I think it's quite sensible, but if we take that to be theoretical leg of the first half of the above contradiction, shouldn't the second half (the one about selfish genetic elements) be expressed in the same terms? I was therefore surprised that the authors did not make more of Gardner & Ubeda's (2017) attempt to describe genomic conflicts using fitness maximizing models. I'm ready to be convinced that I'm wrong here, but I think it would be worth to at least discuss the connection between the modeling approaches.

More seriously, I'm not sure what the novel contribution of the paper is. One of the key results of the paper is that an unlinked suppressor of a distorter can easily invade. This has been known since at least the 1980s. For example, Eshel (1984) sets up almost exactly the same question as described above and in the Introduction and then in Eshel (1985) he goes on to provide pretty much the same solution: when there is perfect recombination between the distorter and its suppressor, the suppressor will be selected for. The link between these kinds of analysis to Leigh's parliament of genes has also been made before (e.g. Ubeda 2006). None of these papers are discussed, or even cited.

Similarly, the authors assume that the number of genes where suppression is favored will greatly outnumber the distorters. One implication of the assumption is that, it is very likely that a suppressor arises before the distorter fixes in the population. I wonder if this is a sensible assumption? The authors cite the fact that distorters are often only revealed in hybrids as evidence of suppression. While this observation is indeed consistent with widespread suppression, it also consistent with distorters simply rapidly spreading to fixation. This is a scenario that, if I understand things correctly, this assumption prevents their models from addressing. Empirically, there are several instances of selfish genetic elements spreading to fixation before suppressors catch up, the P element in *Drosophila* being perhaps the best characterized examples. As the authors point out, the difference in number between distorters and potential suppressors is also Leigh's argument for what prevents "cabals of a few" from taking over the parliament. However as far as I know no one has formally addressed this mathematically. This could have been a really interesting novel contribution.

Overall, I am sorry I cannot be more positive at this time. As I say above, I really like the premise of the paper, but, at least as currently written, I do not see enough new insights to add anything substantial to our understanding of this question.

MINOR COMMENTS AND TYPOS

Line 34 Not all 'selfish genes' in the original/Dawkinsian sense of the term are selfish genetic element. Change to clarify.

Lines 39-42 This sentence is very similar to the last sentence of the previous paragraph. Rephrase or remove.

Line 84 Maybe it's just me, but I did it find a little confusing that the general term chosen was "distorter", which is so similar to segregation distorters like the X chromosome drive system modelled.

Line 97 Is coreplicon really the right word here?

Line 372 Insert WITH between "mates" and "two males".

Line 452-453 With some tweaks, this model could probably provide some nice insights to transposable element proliferation. Lots of data on their suppression

REFERENCES

Eshel, I. 1984. Are Intragametic Conflicts Common in Nature? Do They Represent an Important Factor in Evolution? *Journal of Theoretical Biology*. 108: 159-162

Eshel, I. 1985. Evolutionary Genetic Stability of Mendelian Segregation and the Role of Free Recombination in the Chromosomal System. *American Naturalist*. 125, 412-420

Gardner, A., & Úbeda, F. 2017. The meaning of intragenomic conflict. *Nature Ecology & Evolution*. 1: 1807.

Ubeda, F. 2006. Why Mendelian segregation? *Biochemical Society Transactions*. 34: 566-568.

Reviewer #2:

Remarks to the Author:

This is a nice, thoughtful and very thorough (apart from one exception – see below) modelling exercise on selfish genetic elements.

Much of the debate of the 'parliament' is evoking the "one versus many genes" argument (lines 44-53). My first, rather minor, comment is that the authors (line 55) say that there's many potential problems about this. But much of what it listed on lines 57-65 actually is better described as "tasks for a modeller" than problems, at least if one interprets a problem to mean a potential flaw. For example, the fact that the immediate dynamics depends on the current frequency of a selfish genetic element is surely true, but why is it a 'problem'? A reader would be better informed if one put these tasks in context: what do we already know about them based on previous theory, and what is now new in this MS?

My main comment again relates to the one versus many genes in the following way: the authors don't really fully address its complexities. They consider, throughout the MS, what happens to a pre-existing (though perhaps newly arisen & therefore rare) selfish genetic element. There's often a rather deterministic implicit view, exemplified on line 184, that when a suppressor is needed, then it also

spreads. But this spread actually requires two things: that a suppressor spreads from rare – which was considered and also that it arises in the first place – which was not, except verbally: the authors justify this on lines 489-491, with the usual many genes argument.

However, here a devil's advocate should really remind of the following. Yes, there's many potential genes to suppress a selfish one, but isn't the large number of genes per genome also an opportunity for many of them to become new hopeful selfish elements? Do these factors cancel out? I'm not aware of anyone having modelled the relevant mutation rates (where one would have to make some assumptions: is it easier to become a novel selfish element or a novel suppressor for a particular already existing drive element? – for me, the specificity of the latter task sounds like a lower per locus rate). I believe this should be at least discussed.

Another comment concerns eqn (2). Here, an individual with a suppressor only pays the cost if the suppressor is expressed (the c_{sup} term isn't applied to all combinations involving x_2 or x_4). That's of course an assumption one can make, but one could also want to know how strongly the 'optimistic' findings – i.e. that selfish elements remain rare – rely on this assumption. It would be an easy one to relax.

Minor comments.

The notation, that quickly switches from categories 0 and 1 and + and sup to a sequence 1,2,3,4, so that even the meaning of the label '1' changes within a sentence, is ... let's say it's not the smoothest choice I've ever seen, from the perspective of the reader trying to remember what is what.

As a whole, I was a bit surprised that the Lindholm et al. 2016 TREE review on the eco&evo dynamics of meiotic drive was not cited. I know some people have principles of always citing original work rather than reviews. However, many of the statements made here are discussed in that work too, and in my view, citing review work can be useful for the reader who is keen to know the most helpful starting point when wanting to gain an overview of a subject. Also, assuming the review was a well balanced one (ideally they always are), it minimizes the risk of cherry-picking references to support one's claims.

Reviewers' comments:

Reviewer #1 (Remarks to the Author):

GENERAL COMMENTS

In this paper Scott and West examines the apparent contradiction between the assumption, common among behavioral ecologists, that organisms should appear designed to maximize their (inclusive) fitness and the presence of selfish genetic elements that undermines the organismal 'unity-of-purpose' required for this. The authors develop a series of models of distorters and suppressors to argue that selfish genetic elements will either have too weak fitness costs to be important, or, if the fitness costs to the individual organism are large, they will be readily suppressed.

There is a lot to like about this paper. The contradiction highlighted is an important and understudied one, the text is clearly written and the argument is easy to follow, and (to the best of my ability) the math seems correct. I also appreciated that they adapted their models to three different biological examples. Yet, I am not convinced that the authors actually provide any new answers the stated contradiction.

- **We thank the referee for their kind words. We have added in new work, of the sort that they suggest later in their review, to significantly increase the amount of ‘new answers’.**

To start, the theoretical justification for the design maximization argument is usually found in optimization theory, as established by Alan Grafen and endorsed by the authors. While there are many critics of this approach, I am not one of them. I think it's quite sensible, but if we take that to be theoretical leg of the first half of the above contradiction, shouldn't the second half (the one about selfish genetic elements) be expressed in the same terms? I was therefore surprised that the authors did not make more of Gardner & Ubeda's (2017) attempt to describe genomic conflicts using fitness maximizing models. I'm ready to be convinced that I'm wrong here, but I think it would be worth to at least discuss the connection between the modeling approaches.

- **These are good points, and we see these as complementary approaches. We have added: a link / citation when discussing the evolution of trait distortion; citations at other relevant parts of our paper; and a detailed discussion of the links between Grafen's formal Darwinism, Gardner & Ubeda (2017), and the present paper, to Supplementary Information 9.**
- **As we see it, the aim of Gardner & Ubeda (2017) is to clarify the evolutionary “battleground” over which intra-genomic conflict can play out. If a “battleground” model establishes the causes of intra-genomic conflict, then a “resolution” model addresses the consequences. Our models are resolution models, and complementary to the battleground models described in Gardner & Ubeda (2017). We have discussed this in further detail in Supplementary Information 9.**
- **Most traditional population genetic approaches only allow alleles to adopt discrete strategies. A down side is that the full range of allelic variation is not considered, so the evolutionary equilibria derived from these models might not reflect true long-term evolutionary end points (Eshel 1996). In contrast, the most general forms of our models permit continuous allelic variation in trait distortion and suppression. This allows genes to evolve in the direction of gene fitness maximisation, towards the equilibria derived via optimisation theory in Gardner & Ubeda (2017).**

More seriously, I'm not sure what the novel contribution of the paper is. One of the key results of the paper is that an unlinked suppressor of a distorter can easily invade. This has been known since at least the 1980s. For example, Eshel (1984) sets up almost exactly the same question as described above and in the Introduction and then in Eshel (1985) he goes on to provide pretty much the same solution: when there is perfect recombination between the distorter and its suppressor, the suppressor will be selected for. The link between these kinds of analysis to Leigh's parliament of genes has also been made before (e.g. Ubeda 2006). None of these papers are discussed, or even cited.

- **There are three key differences with Eshel's work. First, Eshel (1985) considered meiotic drivers that drive without systematically biasing organism traits (don't influence trait values). Such drivers may incur deleterious organism effects (costs), arising from linked deleterious genes or from the act of driving itself, but these organismal costs will be disfavoured and countered across the whole genome, including by the meiotic drivers themselves. Such drivers therefore do not compromise organismal design (trait values). You can only examine how much selfish genetic elements will distort traits involved in organismal design if you examine selfish genetic elements that distort traits. Our paper is about is drivers that can influence organismal design - trait distorters that influence trait values.**
- **The second key difference is that we don't just look at whether a trait distorter and suppressor can spread. We look at how trait values are distorted by the interaction between trait distorters and their suppressors. We don't just find that suppressors are favoured. We find that the trait distortion experienced by organisms shows a domed relationship with strength of the trait distorter, and that suppression is most likely when there are larger influences on organismal design.**
- **The third key difference is that we let the drivers evolve – we find that they evolve into the parameter space where they are more likely to be suppressed. Consequently, we suggest that trait distorters will drive their own demise.**
- **The population genetics of simple meiotic drivers, as discussed by Eshel (1985), is different to the population genetics of trait distorters, discussed here. Our illustrative models focus on a hypothetical gene that can only drive at meiosis if an organism trait is distorted.**
- **The comments of the referee have made clear that we need to make these differences clearer. To do this, we have: (i) clarified the need to examine the second and third differences mentioned above in the introduction (paragraph on potential issues with parliament of genes hypothesis); (ii) added a discussion of Eshel (1984) and Eshel (1985), and especially trait distorters versus meiotic drivers, to Supplementary Information 10, and to the Discussion in the main text (lines 561-571); (iii) appropriately cited Úbeda (2006) in the main text.**

Similarly, the authors assume that the number of genes where suppression is favored will greatly outnumber the distorters. One implication of the assumption is that, it is very likely that a suppressor arises before the distorter fixes in the population. I wonder if this is a sensible assumption? The authors cite the fact that distorters are often only revealed in hybrids as evidence of suppression. While this observation is indeed consistent with widespread suppression, it also consistent with distorters simply rapidly spreading to fixation. This is a scenario that, if I understand things correctly, this assumption prevents their models from addressing. Empirically, there are several instances of selfish genetic elements spreading to fixation before suppressors catch up, the P element in *Drosophila* being perhaps the best characterized examples.

- **These are good points, to which we have two main responses. First, we have carried out additional analyses where we relax our assumption that the number of genes where suppression is favoured will greatly outnumber the distorters. We investigated the specific influence of ‘cabal’ size (described in more detail below). This is now a whole new major section of our paper.**
- **Second, there is a key difference between meiotic drivers and the trait distorters that we are interested in. Selfish genetic elements that drive without systematically distorting organism traits, such as P elements and the meiotic drivers discussed by Eshel (1985), will not favour suppression if they reach fixation before the suppressors catch up.**
- **We considered trait distorters. The population genetics is different, meaning that, because organism traits are still distorted once the trait distorter reaches fixation, suppression is still favoured. Our models therefore do not rely on the assumption that the suppressor arises before the trait distorter has reached fixation. Our derivation of the suppressor invasion condition in Appendix 2 in fact assumes that the trait distorter is at fixation.**
- **With respect to hybridization, it is true that, for selfish genetic elements that do not distort organism traits, like simple meiotic drivers, there is no way of distinguishing between selfish genetic elements at fixation and selfish genetic elements under suppression (Corbett-Detig *et al.* 2019). However, for trait distorters, hybridization reveals trait distorters under suppression, because if the hybrids express trait distortion whereas the parents do not, the trait distorters must have been under suppression in the parents (Blows *et al.* 1999).**
- **We have clarified these issues by adding: (i) a new analysis section (Dynamics Models); and (ii) specific coverage of these issues in the discussion.**

As the authors point out, the difference in number between distorters and potential suppressors is also Leigh's argument for what prevents "cabals of a few" from taking over the parliament. However as far as I know no one has formally addressed this mathematically. This could have been a really interesting novel contribution.

- **We have taken this suggestion on board and developed a series of new models to formally address how the proportional size of a “cabal” within a genome affects the average trait distortion experienced by individuals across evolutionary timescales.**
- **We find, with a combination of analytical and numerical population genetic approaches, and agent-based simulations, that, if trait distorters and suppressors can arise repeatedly across evolutionary time at different loci within the genome, the average trait distortion experienced by organisms across evolutionary time may be affected by: the strength and sophistication of trait distorters available to the cabal, genome size, mutation rate, mutational accessibility of suppressors and trait**

distorters. However, all of these parameters have reduced effect as the proportional cabal size within the genome decreases.

- **For limitingly small cabal size, the results of the previous models, which consider the fate of a single trait distorter, are recovered. This was a very pleasing result, suggesting why the parliament of genes hypothesis could be generally applicable, despite variation in biological details across species.**

Overall, I am sorry I cannot be more positive at this time. As I say above, I really like the premise of the paper, but, at least as currently written, I do not see enough new insights to add anything substantial to our understanding of this question.

- **Hopefully we have addressed this by: (1) clarifying the difference between previous work on meiotic drive and our work on trait distorters; (2) adding in our new dynamical analyses that allowed us to investigate factors such as cabal size, genome size and mutation rate. [See also responses to above comments.]**

MINOR COMMENTS AND TYPOS

Line 34 Not all 'selfish genes' in the original/Dawkinsian sense of the term are selfish genetic element. Change to clarify.

- **Changed to say “selfish genetic element”.**

Lines 39-42 This sentence is very similar to the last sentence of the previous paragraph. Rephrase or remove.

- **Sentence rephrased.**

Line 84 Maybe it's just me, but I did it find a little confusing that the general term chosen was "distorter", which is so similar to segregation distorters like the X chromosome drive system modelled.

- **The general term has been changed from “distorter” to “trait distorter” in all cases.**

Line 97 Is coreplicon really the right word here?

- **The concept of a “coreplicon” is tricky and the best way to define it is not obvious (Haig 2014). For this reason, we have decided to remove the terminology, here and elsewhere in the paper, and instead refer to collections of genes favoring a particular kind of trait distortion, called “cabals”, and collections of genes disfavoring a particular kind of trait distortion, called the “commonwealth”. In both cases, we follow the terms suggested by Leigh.**

Line 372 Insert WITH between "mates" and "two males".

- **Amended.**

Line 452-453 With some tweaks, this model could probably provide some nice insights to transposable element proliferation. Lots of data on their suppression

- **Whilst this is of potential interest, transposons are examples of selfish genetic elements that do not need to systematically bias an organism trait in order to drive. Therefore, transposons do not compromise how traits contribute to individual fitness maximization, so are not within the remit of our paper.**

REFERENCES

Eshel, I. 1984. Are Intragametic Conflicts Common in Nature? Do They Represent an Important Factor in Evolution? *Journal of Theoretical Biology*. 108: 159-162

Eshel, I. 1985. Evolutionary Genetic Stability of Mendelian Segregation and the Role of Free Recombination in the Chromosomal System. *American Naturalist*. 125, 412–420

Gardner, A., & Úbeda, F. 2017. The meaning of intragenomic conflict. *Nature Ecology & Evolution*. 1: 1807.

Ubeda, F. 2006. Why Mendelian segregation? *Biochemical Society Transactions*. 34: 566-568.

Reviewer #2 (Remarks to the Author):

This is a nice, thoughtful and very thorough (apart from one exception – see below) modelling exercise on selfish genetic elements.

- **We thank the referee for their kind words.**

Much of the debate of the 'parliament' is evoking the "one versus many genes" argument (lines 44-53). My first, rather minor, comment is that the authors (line 55) say that there's many potential problems about this. But much of what it listed on lines 57-65 actually is better described as "tasks for a modeller" than problems, at least if one interprets a problem to mean a potential flaw. For example, the fact that the immediate dynamics depends on the current frequency of a selfish genetic element is surely true, but why is it a 'problem'? A reader would be better informed if one put these tasks in context: what do we already know about them based on previous theory, and what is now new in this MS?

- **We have amended this paragraph accordingly, to focus on what we do and don't know based on previous theory, and how this could lead to problems for the applicability of the parliament of genes hypothesis.**

My main comment again relates to the one versus many genes in the following way: the authors don't really fully address its complexities. They consider, throughout the MS, what happens to a pre-existing (though perhaps newly arisen & therefore rare) selfish genetic element. There's often a rather deterministic implicit view, exemplified on line 184, that when a suppressor is needed, then it also spreads. But this spread actually requires two things: that a suppressor spreads from rare – which was considered and also that it arises in the first place - which was not, except verbally: the authors justify this on lines 489-491, with the usual many genes argument.

- **We have addressed the problems/limitations suggested by the referee by adding a whole new series of dynamical analyses on these exact issues. Our new models, which formalise Leigh's "cabals of a few" argument, allow the mutational accessibility of trait distorters and suppressors, mutation rate, and genome size, to vary. All of these factors influence the likelihood that a suppressor (or trait distorter) will arise in a population in a given generation.**
- **Our new models, in their general formulations, are also stochastic. We have therefore relaxed the deterministic assumption of the previous models that a suppressor will be readily available. For certain extreme parameter values, the probability that a suppressor arises in a given generation in response to a trait distorter might be infinitesimally small.**

However, here a devil's advocate should really remind of the following. Yes, there's many potential genes to suppress a selfish one, but isn't the large number of genes per genome also an opportunity for many of them to become new hopeful selfish elements? Do these factors cancel out? I'm not aware of anyone having modelled the relevant mutation rates (where one would have to make some assumptions: is it easier to become a novel selfish element or a novel suppressor for a particular already existing drive element? – for me, the specificity of the latter task sounds like a lower per locus rate). I believe this should be at least discussed.

- **This are very good points, which we have addressed in our new dynamical analyses.**
- **One outcome of our analyses is that, if the region of the genome at which trait distortion is favoured is small (proportionally small cabal), then trait distorters will arise slowly over time, even in large genomes. A given locus cannot give rise to a suppressor or a trait distorter. Rather, it can only give rise to a trait distorter if it lies within the cabal; otherwise, it can only give rise to a suppressor.**

Another comment concerns eqn (2). Here, an individual with a suppressor only pays the cost if the suppressor is expressed (the c_{sup} term isn't applied to all combinations involving x_2 or x_4). That's of course an assumption one can make, but one could also want to know how strongly the 'optimistic' findings – i.e. that selfish

elements remain rare – rely on this assumption. It would be an easy one to relax.

- **We have undertaken this analysis and found that the results are qualitatively unchanged when the suppressor is obligately expressed. This has been referenced in the main text (lines 178-180, 229-233, 593-599) and the full analysis of this case is given in Supplementary Information 7.**

Minor comments.

The notation, that quickly switches from categories 0 and 1 and + and sup to a sequence 1,2,3,4, so that even the meaning of the label ‘1’ changes within a sentence, is ... let’s say it’s not the smoothest choice I’ve ever seen, from the perspective of the reader trying to remember what is what.

- **Notation has been changed for clarity.**

As a whole, I was a bit surprised that the Lindholm et al. 2016 TREE review on the eco&evo dynamics of meiotic drive was not cited. I know some people have principles of always citing original work rather than reviews. However, many of the statements made here are discussed in that work too, and in my view, citing review work can be useful for the reader who is keen to know the most helpful starting point when wanting to gain an overview of a subject. Also, assuming the review was a well balanced one (ideally they always are), it minimizes the risk of cherry-picking references to support ones claims.

- **We have added a citation in appropriate places in the main text.**

References

Blows, M.W., Berrigan, D. & Gilchrist, G.W. (1999). Rapid evolution towards equal sex ratios in a system with heterogamety. *Evol Ecol Res*, 277–283.

Corbett-Detig, R., Medina, P., Frérot, H., Blassiau, C. & Castric, V. (2019). Bulk pollen sequencing reveals rapid evolution of segregation distortion in the male germline of Arabidopsis hybrids. *Evolution Letters*, 129, 1393–11.

Eshel, I. (1996). On the changing concept of evolutionary population stability as a reflection of a changing point of view in the quantitative theory of evolution. *Journal of Mathematical Biology*, 34, 1–26.

Haig, D. (2014). Genetic dissent and individual compromise. *Biol Philos*, 29, 233–239.

Reviewers' Comments:

Reviewer #1:

Remarks to the Author:

I was pleased to see the extensive revisions that the authors undertook in response to the comments. I think they have greatly strengthened the paper. Even though (or perhaps because) I don't agree with all their conclusions it has given me a lot to think about and I look forward to the discussions that the paper will generate.

Reviewer #2:

Remarks to the Author:

To a large extent, I find this paper much improved, but I remain a little uncomfortable with the way the authors responded to one of my major comments. To recap, I wondered if the results would be robust if one also takes into account that many genes mean that distorters also have large chances to appear, not only that there's lots of potential sites for suppressors to arise in. The write in their response letter

"First, we have carried out additional analyses where we relax our assumption that the number of genes where suppression is favoured will greatly outnumber the distorters. We investigated the specific influence of 'cabal' size (described in more detail below). This is now a whole new major section of our paper."

Basically, they now (in some of their models) assume that the genome is split into 2 chunks - potential 'cabal' sites and the rest is then 'commonwealth'; they consider 'cabal' sizes up to half of the genome, and also point out that empirically found 'cabals' are small.

Once again, I find these to be assumptions that may bias the results towards a particular result (that distortion remains rare or mild). The view I was trying to convey is that there is probably nothing that prevents, in principle, any site in a genome from becoming a distorter, or at least that appears to be a sensible null assumption. If it indeed turns into a distorter, then in a post hoc manner we will assign that region the role of a cabal, and the rest is commonwealth. The fact that realized cabals are small does not constitute evidence that only a small part of a genome is in principle able to mutate to be a new distorter.

If one makes a priori an assumption that only specific parts of a genome have any pre-existing ability to become 'rebellious' then indeed the results hold - but I don't really buy the arguments presented here why that should be the case, since it appears to conflate small genomic sizes of existing examples with the existence of a constraint that these sites are the only places where such examples could have arisen, even in principle. My suggestion in the previous round, to examine the issue via mutation, would go as follows: any site can become a distorter (turning part of the genome into a cabal region if one wants to use that terminology), and any site can also become a suppressor. The mutation rates from 'normal' to 'distorter' and from 'normal' to 'suppressor' can be unequal, and I wouldn't a priori constrain the analysis to one being larger than the other.

My minor comments are all about some awkward formulations in the abstract/introduction:

Line 12. Why the plural - I think you've only specified one contradiction.

Lines 29-31. It would be nice if the text consisted of sentences (a sentence always has a verb).

Line 33. When one already says 'on the other hand' then 'however' seems superfluous.

Line 40. Make the full stop a comma, and the preceding sentence starts working. (Right now the sentence 'The contradiction is...' stops before it's managed to describe the bit that actually makes it a contradiction.) I would also remove the last sentence of this paragraph, it's getting repetitive to have the contradiction there so many times, especially when the next para goes on about it again.

Line 58. In the previous round of review I wrote that the authors seem to exaggerate the problems with this body of theory, and I still think 'undermine' is a too strong word choice here.

Line 63-65. Something about the logic of this sentence does not work. The 'even if' structure seems to conflict with the content of what follows.

Reviewers' comments:

Reviewer #1 (Remarks to the Author):

I was pleased to see the extensive revisions that the authors undertook in response to the comments. I think they have greatly strengthened the paper. Even though (or perhaps because) I don't agree with all their conclusions it has given me a lot to think about and I look forward to the discussions that the paper will generate.

Thank you.

Reviewer #2 (Remarks to the Author):

To a large extent, I find this paper much improved, but I remain a little uncomfortable with the way the authors responded to one of my major comments. To recap, I wondered if the results would be robust if one also takes into account that many genes mean that distorters also have large chances to appear, not only that there's lots of potential sites for suppressors to arise in. The write in their response letter "First, we have carried out additional analyses where we relax our assumption that the number of genes where suppression is favoured will greatly outnumber the distorters. We investigated the specific influence of 'cabal' size (described in more detail below). This is now a whole new major section of our paper."

Basically, they now (in some of their models) assume that the genome is split into 2 chunks - potential 'cabal' sites and the rest is then 'commonwealth'; they consider 'cabal' sizes up to half of the genome, and also point out that empirically found 'cabals' are small.

Once again, I find these to be assumptions that may bias the results towards a particular result (that distortion remains rare or mild). The view I was trying to convey is that there is probably nothing that prevents, in principle, any site in a genome from becoming a distorter, or at least that appears to be a sensible null assumption. If it indeed turns into a distorter, then in a post hoc manner we will assign that region the role of a cabal, and the rest is commonwealth. The fact that realized cabals are small does not constitute evidence that only a small part of a genome is in principle able to mutate to be a new distorter.

If one makes a priori an assumption that only specific parts of a genome have any pre-existing ability to become 'rebellious' then indeed the results hold – but I don't really buy the arguments presented here why that should be the case, since it appears to conflate small genomic sizes of existing examples with the existence of a constraint that these sites are the only places where such examples could have arisen, even in principle.

My suggestion in the previous round, to examine the issue via mutation, would go as follows: any site can become a distorter (turning part of the genome into a cabal region if one wants to use that terminology), and any site can also become a suppressor. The mutation rates from 'normal' to 'distorter' and from 'normal' to 'suppressor' can be unequal, and I wouldn't a priori constrain the analysis to one being larger than the other.

We apologise for not completely addressing this issue in our previous revision. We had misunderstood exactly what the referee had been suggesting. This raises a number of interesting issues, which we now address in our paper.

We have now added in the suggested model – it gives the same results as our previous dynamic models and does not alter our conclusions. We have added this to Supplementary Information 12.

However, we disagree with the rationale for this model. We suspect that the suggestion for this model arises from a misunderstanding of our purpose and scope of our models. This made us realise some issues that we needed to better explain. We have addressed this: (1) with clarification in the main text (lines 151-165, 379-391, 422-433); and (2) by adding in a new section in the supplementary information (Supplementary Information 12). We will now summarise the key issues here:

We posit that, for any given kind of trait distortion, we can divide the genome into two sections: the cabal, comprising those loci where the trait distortion is favoured, and the commonwealth, comprising those loci where the trait distortion is disfavoured. The reviewer takes issue with us defining the cabal and commonwealth ‘a priori’, and argues instead that we should allow trait distorters and suppressors to arise at any locus in a genome. The cabal and commonwealth could then be defined ‘ad hoc’ if we wish, and these aggregations would presumably grow and shrink as new trait distorters and suppressors mutate in and out at different loci.

Justification of modelling assumptions

In Supplementary Information 12, we have added a detailed description of how our work relates to Cosmides & Tooby (1981), which showed that genomes can be divided up into ‘coreplicons’, which are groups of loci that are inherited in the same way, so share the same maximand. Coreplicons are defined *a priori* based on shared inheritance, not on the observation, empirically or in the context of a theoretical model, of trait-affecting loci (Grafen, 2006; Burt & Trivers, 2006; Haig, 2014).

We argue in Supplementary Information 12 that ‘cabals’ and ‘commonwealths’ should also be defined in this way, *a priori*. The cabal comprises all coreplicons that favour a particular kind of trait distortion away from individual fitness maximisation, and the commonwealth comprises the remaining coreplicons. It is useful, when analysing a particular trait, to partition and sum up coreplicons in this way, because the resolution of this conflict – between the cabal and commonwealth – informs how far a given trait should deviate from individual fitness maximisation.

Our models are meant to address whether selfish genetic elements can distort organism traits away from individual fitness maximisation, where the ‘individual’ here really means the majority interest within the parliament of genes (Grafen, 2006). This is why we only considered cabal sizes of up to a half. If the cabal was greater than half of the genome, it would reflect the majority interest within the parliament, so would cease to be a cabal. Our models therefore consider the full range of scenarios depicting potential distortion of an organism trait from individual fitness maximisation.

Modelling extension

Having said all this, and having justified our approach, which defines the cabal and commonwealth *a priori*, we now undertake the theoretical exercise asked for by the reviewer, teasing out the biological assumptions required for this scenario to make sense, and arguing that this scenario: (i) is biologically implausible, and (ii) leads to the same results as our previous models.

The reviewer states that: “there is probably nothing that prevents, in principle, any site in a genome from becoming a distorter, or at least that appears to be a sensible null assumption.”

Our first point is that, contrary to this statement, most sites in the genomes of biological organisms *cannot* become trait distorters. Most loci in a genome are unimprinted, vertically inherited and autosomal. Therefore, for an organism approximating individual fitness maximisation, no conceivable distortion of an organism trait could possibly give these loci a propagation advantage. Of course, meiotic drivers or transposons could arise at any of these loci, and the resulting selfish genetic elements could spread through the population as a result. However, *trait distorters* could not arise at these loci – the transmission of alleles at these loci is maximised when the organism trait values are those which lead to individual fitness maximisation (Grafen, 2006; Haig, 2014). The key difference here is between meiotic drive (favoured at any locus; selfish benefit does not arrive via distorting a trait) and selfish genetic elements that gain a benefit by distorting a trait (e.g. the specific examples that we model).

Nevertheless, we will imagine a hypothetical organism where any site in its genome could give rise to a trait distorter. The question then becomes: what type of trait distortion is favoured at each locus? It could firstly be the case that each locus gains its selfish propagation advantage by distorting a unique trait, or by distorting a common trait but along a unique dimension (axis) and direction. If this is the case, each locus in the genome would effectively form its own cabal, with a proportional size within the genome approximating zero ($\theta \rightarrow 0$). It could alternatively be the case that groups of loci favour the

same type of trait distortion (same trait, dimension and direction), meaning proportional cabal sizes can be greater ($\theta > 0$). However, given that the size of any one cabal cannot exceed a half (else that group of loci would cease to be a cabal), it must logically be the case that (at least two) different types of trait distortion are favoured across the genome.

We now assume that the rate of trait distorter introduction, per generation, per locus, in some genome within the population, is ρ_{D1} . We take the number of loci within a genome to be γ , which means that new trait distorters are introduced into the population every $1/(\rho_{D1}\gamma)$ generations. This is a faster rate than previously considered in our Dynamics models, which was dependent on proportional cabal size ($1/(\theta\rho_{D1}\gamma)$). As was the case in our Dynamics models, the suppressor of a given trait distorter will be expected to arise after a lag of $(1/(1-\theta)\rho_{S1}\gamma)$ generations, where ρ_{S1} is the rate of suppressor introduction, per generation, per locus, for any locus situated outside of the target trait distorter's cabal.

So in this new theoretical scenario, compared to our previous Dynamics models, trait distorters are arising at a faster rate, but they are suppressed at the same rate as before. This would apparently suggest that average trait distortion should be more appreciable in this new scenario. However, this is not the case. The rate that trait distorters *that distort a given trait* are introduced is the same as our Dynamics models ($1/(\theta\rho_{D1}\gamma)$). This new formulation appears to favour increased deviation of organisms from individual fitness maximisation, but this is not the case, as the new scenario is implicitly considering the distortion of multiple traits simultaneously. The distortion of any *given trait* from individual fitness maximisation is still accurately given by our Dynamics models.

Therefore, the scenario highlighted by the reviewer implicitly refers to a scenario where multiple traits are being distorted and restored simultaneously, in the context of a single model. However, there is no reason why the evolution of distortion and suppression at one trait should be affected by the evolution of distortion and suppression at any other trait. This is why the results of the new theoretical scenario converge on our Dynamics models once we consider a single type of trait distortion in isolation. As mentioned earlier, we believe that our models cover the full range of scenarios depicting potential distortion of an organism trait from individual fitness maximisation. The modelling extension, as well as being biologically implausible, provides no additional insight.

To conclude, we think that there was some confusion, where both we and the referee misunderstood each other. This made clear some issues that we needed to clarify in our paper, which we have now done, both in the main text, and by adding in a new section to the supplementary information (Supplementary Information 12). We thank the referee for making the point that enabled this, to the improvement and generality of our paper.

My minor comments are all about some awkward formulations in the abstract/introduction:

Line 12. Why the plural – I think you've only specified one contradiction.

Amended.

Lines 29-31. It would be nice if the text consisted of sentences (a sentence always has a verb).

Amended.

Line 33. When one already says 'on the other hand' then 'however' seems superfluous.

Amended.

Line 40. Make the full stop a comma, and the preceding sentence starts working. (Right now the sentence 'The contradiction is...' stops before it's managed to describe the bit that actually makes it a contradiction.) I would also remove the last sentence of this paragraph, it's getting repetitive to have the contradiction there so many times, especially when the next para goes on about it again.

Amended.

Line 58. In the previous round of review I wrote that the authors seem to exaggerate the problems with this body of theory, and I still think 'undermine' is a too strong word choice here.

'Undermine' has been changed to 'affect'.

Line 63-65. Something about the logic of this sentence does not work. The 'even if' structure seems to conflict with the content of what follows.

Amended for clarity.

References

Burt, A. & Trivers, R. 2006. *Genes in Conflict*. Harvard University Press, Cambridge, MA and London, England.

Cosmides, L.M. & Tooby, J. 1981. Cytoplasmic inheritance and intragenomic conflict. *Journal of Theoretical Biology* 89: 83–129.

Grafen, A. 2006. Optimization of inclusive fitness. *Journal of Theoretical Biology* 238: 541–563.

Haig, D. 2014. Genetic dissent and individual compromise. *Biol Philos* 29: 233–239.

Reviewers' Comments:

Reviewer #2:

Remarks to the Author:

I appreciate the continued effort of the authors (the supplementary file is very very long). I think this is a healthy conversation: modellers need to be aware of whether they are, perhaps inadvertently, looking at best-case scenarios for their argument or perhaps the opposite. I agree now that my "any locus" argument is, in turn, a bit of a worst-case argument. But I hope that the authors can see that there remains a more refined version of a worry. Namely: Even if one does not want to think about the "any locus can turn rebellious" case, the fact remains that the predefined potential-cabal identity means that the authors assume that there's just one of such potential candidate anywhere in the genome. The rest is a priori assumed to be consisting of commonwealth only. I find this to be quite a strong assumption, and my hunch is that it's shifting the model towards a best-case scenario (compared to an open investigation of all possible scenarios). After all, the argument that focus on a 'given trait' (in S12) could be contested. If the question is "why are distorters rare?" then we're not a priori defining the question as "rare for THIS trait", but generally interested in why they're so hard to find (for any trait).

At the same time I don't want to prolong the model modifications forever, as I believe the contribution is worthwhile and already really quite big. But if the authors wanted to modify the text and consider exposing this assumption, it'd make the final product more trustworthy, in my eyes at least.

REVIEWERS' COMMENTS:

Reviewer #2 (Remarks to the Author):

I appreciate the continued effort of the authors (the supplementary file is very very long). I think this is a healthy conversation: modellers need to be aware of whether they are, perhaps inadvertently, looking at best-case scenarios for their argument or perhaps the opposite. I agree now that my "any locus" argument is, in turn, a bit of a worst-case argument. But I hope that the authors can see that there remains a more refined version of a worry. Namely: Even if one does not want to think about the "any locus can turn rebellious" case, the fact remains that the predefined potential-cabal identity means that the authors assume that there's just one of such potential candidate anywhere in the genome. The rest is a priori assumed to be consisting of commonwealth only. I find this to be quite a strong assumption, and my hunch is that it's shifting the model towards a best-case scenario (compared to an open investigation of all possible scenarios). After all, the argument that focus on a 'given trait' (in S12) could be contested. If the question is "why are distorters rare?" then we're not a priori defining the question as "rare for THIS trait", but generally interested in why they're so hard to find (for any trait).

At the same time I don't want to prolong the model modifications forever, as I believe the contribution is worthwhile and already really quite big. But if the authors wanted to modify the text and consider exposing this assumption, it'd make the final product more trustworthy, in my eyes at least.

Thank you.

Our analysis has been exclusively focused on single traits in isolation (sex ratio; altruism; public goods production). We have addressed how far selfish genetic elements distort traits from the trait values where individual fitness maximisation is achieved. The reviewer points out that, were we to consider the organism as a whole, at once, and model all organism traits, we could begin to analyse organismal deviation from individual fitness maximisation in N-dimensional space, with respect to N traits simultaneously.

The reviewer is right to point out that we are only showing that trait distortion is negligible for "this (one) trait". We are not explicitly showing that trait distortion is negligible for the organism as an N-dimensional whole. However, we believe that the two notions coincide, given that the evolution of distortion at one trait is likely to proceed independently of the evolution of distortion at any other trait. We therefore have chosen to leave the models as they are. Modelling traits in isolation has the advantage of analytical tractability as well as conceptual clarity.

Having said that, we have added a section to the supplementary information (Supplementary Discussion 1, under the "Individual fitness maximisation: emergence versus maintenance" subheading) and a sentence to the main text (lines 601-604) to expose a related modelling assumption – namely that, given that cabals and commonwealths have access to different types of allele (trait distorters and suppressors, respectively) there is a bias, meaning our models deal with the *maintenance* rather than the *emergence* of individual fitness maximisation.